# Joint Geometric and Trajectory
# Consistency Learning for One-Step Real-World Super-Resolution

**Chengyan Deng** [1]  **Zhangquan Chen** [2]  **Li Yu** [† 1 3]  **Kai Zhang** [4]  **Xue Zhou** [3 1]  **Wang Zhang** [1]

## Abstract

Diffusion-based Real-World Image Super-Resolution (Real-ISR) achieves impressive perceptual quality but suffers from high computational costs due to iterative sampling. While recent distillation approaches leveraging large-scale Text-to-Image (T2I) priors have enabled one-step generation, they are typically hindered by prohibitive parameter counts and the inherent capability bounds imposed by teacher models. As a lightweight alternative, Consistency Models offer efficient inference but struggle with two critical limitations: the accumulation of consistency drift inherent to transitive training, and a phenomenon we term "Geometric Decoupling"— where the generative trajectory achieves pixel-wise alignment yet fails to preserve structural coherence. To address these challenges, we propose GTASR (Geometric Trajectory Alignment Super-Resolution), a simple yet effective consistency training paradigm for Real-ISR. Specifically, we introduce a Trajectory Alignment (TA) strategy to rectify the tangent vector field via full-path projection, and a Dual-Reference Structural Rectification (DRSR) mechanism to enforce strict structural constraints. Extensive experiments verify that GTASR delivers superior performance over representative baselines while maintaining minimal latency. The code and model will be released at https://github.com/Blazedengcy/GTASR.

[1]School of Automation Engineering, University of Electronic Science and Technology of China, Chengdu, China [2]Tsinghua Shenzhen International Graduate School, Tsinghua University, Shenzhen, China [3]Shenzhen Institute for Advanced Study, University of Electronic Science and Technology of China, Shenzhen, China [4]School of Intelligence Science and Technology, Nanjing University, Suzhou, China. Correspondence to: Li Yu[†] <lyu@uestc.edu.cn>.

*Proceedings of the 43rd International Conference on Machine Learning*, Seoul, South Korea. PMLR 306, 2026. Copyright 2026 by the author(s).

## 1. Introduction

Image Super-Resolution (ISR) aims to reconstruct a High-Resolution (HR) image from its corresponding degraded Low-Resolution (LR) input. Traditional ISR methods (Liang et al., 2021; Deng et al., 2026b) are often limited to handling known degradation types and struggle to address the complex and unknown degradations encountered in real-world scenarios. Consequently, recent research (Lin et al., 2024; Chen et al., 2025) has gradually shifted its focus to Real-World Super-Resolution (Real-ISR), demonstrating superior practical utility.

Early Real-ISR methods primarily employed Generative Adversarial Networks (GANs) (Goodfellow et al., 2014; Radford et al., 2015). While capable of generating rich textures and realistic visual effects, they are often plagued by training instability and artifacts. In recent years, diffusion models (Ho et al., 2020; Rombach et al., 2022; Yu et al., 2025b;a) have demonstrated immense potential due to their powerful capability in modeling complex distributions. However, their inherent iterative denoising nature leads to an extremely slow inference process. To achieve GAN-level efficiency, recent studies (Yue et al., 2023; Zhang et al., 2025) compress sampling to 10 steps by optimizing the diffusion process or noise injection schemes.

Furthermore, some studies leverage distillation techniques to achieve one-step super-resolution. Early distillation-based attempts like SinSR (Wang et al., 2024b) accelerated inference but struggled with generation quality. To address this, subsequent works shifted towards distilling large-scale Text-to-Image (T2I) priors (Podell et al., 2023; Labs, 2023) to construct stronger student models (Wu et al., 2024a; Sun et al., 2025). However, the massive parameter count of T2I models severely restricts the feasibility of lightweight deployment, while the generative capability of the student model is inherently constrained by the teacher model. In light of this, new generative paradigms are gradually gaining attention. Recently, CTMSR (You et al., 2025) introduced Consistency Training (CT) (Song et al., 2023) and Distribution Trajectory Matching (DTM) into the SR task, achieving one-step SR without distillation. Despite achieving promising results, we observe that CTMSR still suffers from two critical limitations: (i) Insufficient consistency preservation.

The inherent transitive learning mechanism of CT leads to inevitable error accumulation; (ii) Geometric Decoupling. Although DTM improves perceptual quality significantly, it lacks explicit structural constraints, leading to incoherent geometric structures and sub-optimal texture recovery.

In this paper, we propose a novel method called GTASR (Geometric Trajectory Alignment Super-Resolution) to achieve one-step SR by jointly rectifying the noising trajectory and geometric structure. Specifically, GTASR consists of two components: Trajectory Alignment (TA) and Dual-Reference Structural Rectification (DRSR). TA incorporates a full-path projection strategy to mitigate the consistency drift inherent in standard CT strategy. By projecting predictions back onto the noisy manifold to align with the ground-truth, this mechanism explicitly rectifies the tangent vector field to ensure accurate evolution directions. DRSR is designed to bridge the geometric consistency gap where pixel-wise alignment fails to preserve structural fidelity. This module enforces structural constraints by leveraging guidance from both the real trajectory and ground truth to effectively recover high-frequency details.

- We present GTASR, a one-step super-resolution method that introduces a simple yet effective innovation to the Consistency Training paradigm.

- We propose a Trajectory Alignment (TA) strategy that aligns intermediate states with target states to prevent error accumulation. Additionally, we design a Dual-Reference Structural Rectification (DRSR) module that leverages structural guidance to effectively restore high-frequency details.

- Extensive empirical evaluations demonstrate the effectiveness of GTASR across both synthetic and real-world benchmarks, achieving superior realism and structural integrity.

## 2. Related Work

### 2.1. Image Super-Resolution

Early deep learning-based image SR methods (Zhou et al., 2023; Long et al., 2025; Lee et al., 2025) primarily focused on improving fidelity metrics such as Peak Signal-to-Noise Ratio (PSNR) and Structural Similarity (SSIM). Although these works achieved high objective scores through improved network architectures and attention mechanisms, their reconstruction results often appear overly smooth and lack high-frequency details, as their optimization objectives are based on pixel-wise distances (e.g., L1 loss). Furthermore, these methods mostly rely on simple, predefined degradation models, limiting their effectiveness in handling complex real-world degradations. To address these limitations, generative models (Wu et al., 2025a; Li et al.,

2025; Fei et al., 2025; Deng et al., 2026a) have been increasingly applied to real-world SR. Pioneering works like BSRGAN (Zhang et al., 2021) and Real-ESRGAN (Wang et al., 2021) introduced complex degradation operators to simulate real-world scenarios, significantly enhancing perceptual quality; however, GAN-based models often suffer from training instability and visual artifacts. In contrast, diffusion-based methods have proven more effective in improving perceptual quality. Recent studies (Yu et al., 2024; Wu et al., 2024b) leverage the powerful priors of T2I models to achieve exceptional realism. Nevertheless, these methods face prohibitive computational costs and slow inference speeds due to the iterative denoising nature that typically requires tens to hundreds of sampling steps, restricting their practical application.

### 2.2. Acceleration of Diffusion-based Image Super-Resolution

To address the high computational overhead of T2I models in SR tasks caused by multi-step inference, research has shifted toward acceleration strategies, covering sampling process optimization, knowledge distillation, and novel generative paradigms. Regarding sampling optimization, ResShift (Yue et al., 2023) reduced inference to 15 steps by adjusting the noise starting point, while UPSR (Zhang et al., 2025) further compressed the process to 4 steps using a partitioned noise strategy. To further eliminate the iterative cost and enable one-step generation, distillation techniques have been widely adopted. For instance, SinSR (Wang et al., 2024b) utilized distillation to accelerate ResShift, while OSEDiff (Wu et al., 2024a) achieved one-step SR by distilling priors from Stable Diffusion model (Blattmann et al., 2023) via VSD loss (Wang et al., 2023b). However, OSEDiff's performance is bounded by the teacher model, and its massive parameter count impedes practical deployment. Recently, CTMSR (You et al., 2025) leveraged the characteristics of Consistency Training to achieve efficient one-step SR without teacher supervision, demonstrating immense potential. Nonetheless, CTMSR still faces limitations in ensuring recovering fine texture details, leaving its potential for one-step generation not fully exploited.

## 3. Method

As shown in Fig. 1, the proposed GTASR framework adopts a progressive two-stage training strategy. We first outline the preliminaries, followed by our method formulation.

### 3.1. Preliminary

**Diffusion Model.** Standard diffusion models recover data from pure Gaussian noise via a reverse Markov chain. ResShift (Yue et al., 2023) introduces a novel Markov chain tailored for image SR by initializing the reverse process

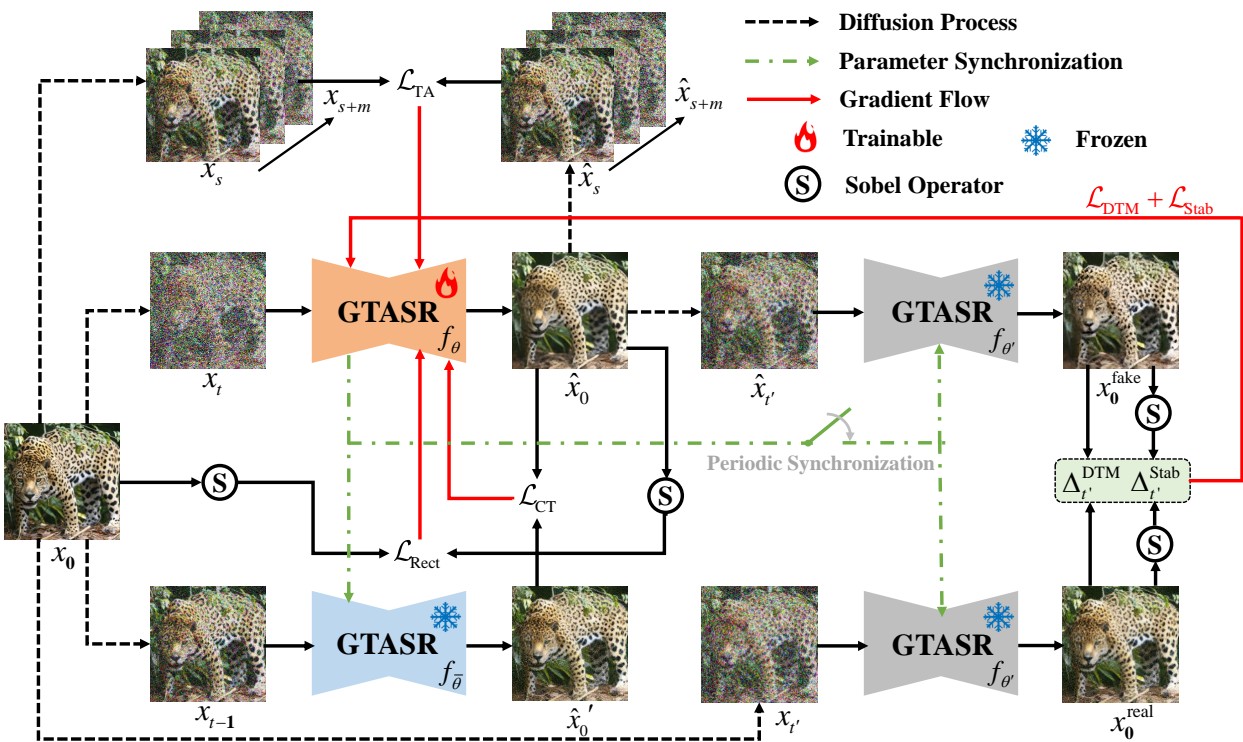

*Figure 1.* The pipeline of the proposed GTASR. We employ a two-stage training scheme. In Stage I, we train the online model $f_\theta$ using the standard $\mathcal{L}_{\text{CT}}$, augmented by our proposed $\mathcal{L}_{\text{TA}}$, where the reference parameters $\theta^-$ are updated at every step using the stop-gradient online parameters. In Stage II, we address geometric decoupling by introducing a target model $f_{\theta'}$ initialized with the pre-trained weights from Stage I, with its parameters updated via periodic synchronization to ensure temporal alignment. We feed the intermediate states $\hat{x}_{t'}$ and $x_{t'}$ into $f_{\theta'}$ to obtain the trajectory endpoints ($x_0^{\text{fake}}$ and $x_0^{\text{real}}$), which are directly utilized for $\mathcal{L}_{\text{DTM}}$ while their structure maps extracted via the Sobel operator are utilized for the proposed DRSR objectives ($\mathcal{L}_{\text{Stab}}$ and $\mathcal{L}_{\text{Rect}}$). Finally, the gradients derived from $\mathcal{L}_{\text{CT}}$, $\mathcal{L}_{\text{DTM}}$, $\mathcal{L}_{\text{Stab}}$, and $\mathcal{L}_{\text{Rect}}$ are backpropagated to $f_\theta$ to jointly enhance perceptual realism and structural integrity.

with a noise-perturbed low-resolution (LR) image, explicitly leveraging LR priors. To construct such a transition, the forward diffusion process is formulated as:

$$x_t = \mathcal{Q}(x_0, y_0, t) = x_0 + \alpha_t e_0 + \sigma_t \epsilon, \quad (1)$$

where $x_0$ and $y_0$ denote the high-resolution (HR) and LR images, respectively, while $e_0 = y_0 - x_0$ represents the residual. The term $\epsilon \sim \mathcal{N}(0, I)$ denotes standard Gaussian noise. The coefficients $\alpha_t$ and $\sigma_t$ are defined as monotonically increasing functions of time $t$, satisfying the boundary conditions $\alpha_0 = \sigma_0 = 0$ and $\alpha_T = \sigma_T = 1$.

Following (Karras et al., 2022; Song et al., 2020), we adopt the standard PF-ODE sharing the forward marginals with the Stochastic Differential Equation (SDE) to describe the deterministic trajectory dynamics:

$$dx = [\dot{\alpha}(t)e_\theta(x, y_0, t) + \dot{\sigma}(t)\epsilon_\theta(x, y_0, t)]dt, \quad (2)$$

where $e_\theta$ and $\epsilon_\theta$ represent the residual and noise predicted by the model parameterized by $\theta$, respectively.

**Consistency Training.** Consistency Models (Song et al., 2023) learn a function $f_\theta$ that maps any state $x_t$ on the

PF-ODE trajectory directly to its origin $x_0$. Formally, this function parameterizes the integral solution of the PF-ODE:

$$f_\theta(x_t, t) \approx x_0 = x_t + \int_t^0 \frac{dx_s}{ds}ds. \quad (3)$$

The core principle enforces self-consistency, requiring that any state $x_t$ along the PF-ODE trajectory maps to the identical origin $x_0$. To achieve this, Consistency Training (CT) minimizes the following objective:

$$\mathcal{L}_{\text{CT}} = \mathbb{E}_{x,t}\Big[d_{\text{I}}\big(f_\theta(x_t, y_0, t), f_{\theta^-}(x_{t-1}, y_0, t-1)\big)\Big], \quad (4)$$

where $d_{\text{I}}(\cdot, \cdot)$ is a distance metric and $\theta^- \leftarrow \text{stopgrad}(\theta)$.

**Distribution Trajectory Matching.** To bridge the distributional gap, CTMSR employs Distribution Trajectory Matching (DTM) (You et al., 2025) using a frozen model $f_{\theta'}$ initialized from a model trained with $\mathcal{L}_{\text{CT}}$. Specifically, it aligns the fake and real trajectories derived from $\hat{x}_t' = \mathcal{Q}(\hat{x}_0, y_0, t')$ and $x_t' = \mathcal{Q}(x_0, y_0, t')$, respectively, where $\hat{x}_0 = f_\theta(x_t, y_0, t)$, and the trajectory endpoints $x_0^{\text{fake}}$ and $x_0^{\text{real}}$ are given by the $f_{\theta'}$ predictions at time $t'$ as shown in Figure. 1. The discrepancy between the two trajectories is quantified by $\Delta_{t'}^{\text{DTM}}$, which is computed as:

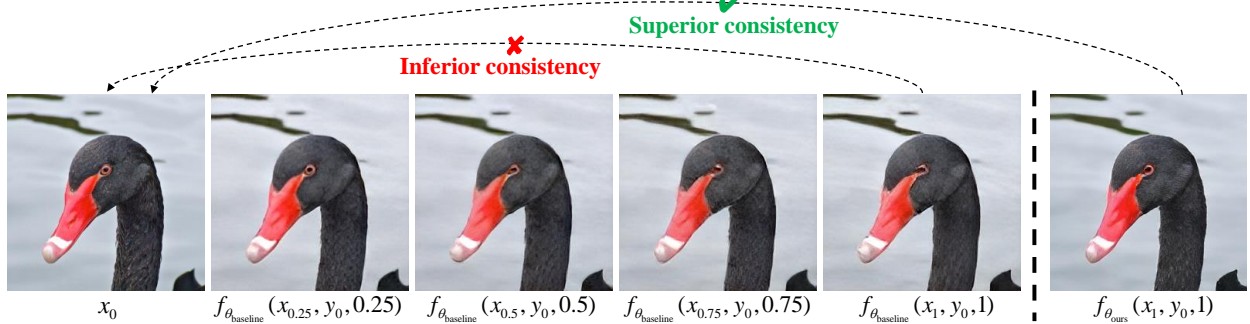

$x_0$     $f_{\theta_{\text{baseline}}}(x_{0.25}, y_0, 0.25)$     $f_{\theta_{\text{baseline}}}(x_{0.5}, y_0, 0.5)$     $f_{\theta_{\text{baseline}}}(x_{0.75}, y_0, 0.75)$     $f_{\theta_{\text{baseline}}}(x_1, y_0, 1)$     $f_{\theta_{\text{ours}}}(x_1, y_0, 1)$

*Figure 2.* Visual analysis of consistency across timesteps. (Left) Baseline CT suffers from severe trajectory drift and detail loss as $t$ increases, particularly in the eye region. (Right) After introducing TA strategy effectively preserves sharp features even at large time steps.

$$\Delta_{t'}^{\text{DTM}} = x_0^{\text{fake}} - x_0^{\text{real}} = f_{\theta'}(\hat{x}_{t'}, y_0, t') - f_{\theta'}(x_{t'}, y_0, t'). \tag{5}$$

To explicitly penalize $\Delta_{t'}^{\text{DTM}}$, DTM loss is formulated as:

$$\mathcal{L}_{\text{DTM}} = \mathbb{E}_{x,t,t'}\Big[\omega(t') \cdot d_{\text{II}}\big(\hat{x}_0, \underbrace{(\hat{x}_0 - \Delta_{t'}^{\text{DTM}})}_{\text{Stop-Gradient}}\big)\Big], \tag{6}$$

where $\omega(t')$ is a time-dependent term to balance gradient magnitudes and $d_{\text{II}}(\cdot, \cdot)$ is a distance metric. More details can be found in the Appendix C.3.

### 3.2. Trajectory Alignment via Full-Path Projection

Although CT enables one-step generation, this often comes at the expense of detail reconstruction performance. As illustrated to the left of the dashed line in Figure 2, the model exhibits inconsistent predictions when trained solely with $\mathcal{L}_{\text{CT}}$. Specifically, while the coarse silhouette remains discernible at large time steps, fine-grained details—particularly the clarity and boundary of the eye — are severely degraded or completely washed out in the noisy state $x_t$ at such intervals. Since $x_t$ retains critical detail cues only at small time steps, high-fidelity reconstruction is inevitably restricted to this narrow regime, leading to the observed trajectory drift and blurred artifacts when consistency supervision is enforced on high-noise states lacking explicit details.

As derived in the Appendix A.2 based on the forward process (Eq. 1), the PF-ODE governing the generation dynamics is formulated as:

$$\frac{\mathrm{d}x_t}{\mathrm{d}t} = v_t^{\text{predict}} = \frac{n}{t}\big(x_t - f_\theta(x_t, y_0, t)\big), \tag{7}$$

where $n$ is determined by the noise schedule described in Appendix C.1. Ideally, the trajectory evolution should be guided by $x_0$, yielding the optimal velocity:

$$\frac{\mathrm{d}x_t}{\mathrm{d}t} = v_t^{\text{ideal}} = \frac{n}{t}(x_t - x_0). \tag{8}$$

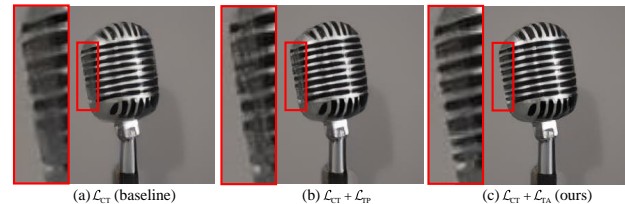

(a) $\mathcal{L}_{\text{CT}}$ (baseline)     (b) $\mathcal{L}_{\text{CT}} + \mathcal{L}_{\text{TP}}$     (c) $\mathcal{L}_{\text{CT}} + \mathcal{L}_{\text{TA}}$ (ours)

*Figure 3.* Visual comparison of different strategies. (a) The baseline $\mathcal{L}_{\text{CT}}$ fails to model fine details effectively. (b) Integrating $\mathcal{L}_{\text{TP}}$ with $\mathcal{L}_{\text{CT}}$ guides the reconstruction direction; however, the metallic grille remains overly blurred and lacks definition. (c) Ours restores intricate mesh textures with superior clarity.

To enforce self-consistency, $\mathcal{L}_{\text{CT}}$ aligns the prediction $f_\theta(x_t, y_0, t)$ with the target derived from the adjacent step $f_{\theta^-}(x_{t-1}, y_0, t-1)$. However, the transition to this adjacent state $x_{t-1}$ is strictly governed by the predicted tangent vector $v_t^{\text{predict}}$. At large timesteps $t$, the inherent uncertainty of $f_\theta(x_t, y_0, t)$ results in an erroneous update direction compared to the ideal Eq. 8. From the perspective of training supervision, this implies that the calculated target $f_{\theta^-}(x_{t-1}, y_0, t-1)$ itself deviates from the optimal path. By minimizing $\mathcal{L}_{\text{CT}}$, the model is forcibly optimized to fit this biased target, inevitably learning an erroneous trajectory that drifts away from the underlying image manifold, as evidenced by Figure 2.

To mitigate this drift, a straightforward strategy is to introduce explicit Ground Truth (GT) supervision as a corrective signal to recalibrate the velocity field. We formulate this as the Target Point Loss:

$$\mathcal{L}_{\text{TP}} = \mathbb{E}_{x,t}\Big[d_{\text{I}}\big(f_\theta(x_t, y_0, t), x_0\big)\Big]. \tag{9}$$

While $\mathcal{L}_{\text{TP}}$ uses GT guidance for directional optimization, its indiscriminate constraints at high noise levels disregard the inherent generative uncertainty. This mismatch perturbs tangent vector estimation, leading to the sub-optimal structure and residual blurring as shown in Figure. 3(b).

Motivated by the intrinsic properties of the forward diffusion process, we address the trade-off between content consis-

tency and detail fidelity. Specifically, the noise schedule acts as an implicit dynamic filter governed by the Signal-to-Noise Ratio (SNR). At large time steps $t$, the dominant noise $\sigma_t$ naturally masks high-frequency reconstruction errors, forcing the optimization to prioritize the global semantic structure (i.e., the correct evolution direction). Conversely, as $t \to 0$ (where $\sigma_t \to 0$), the noise fades, progressively shifting the focus toward high-frequency texture consistency. Driven by this insight, we propose the Trajectory Alignment (TA) strategy. Instead of applying $\mathcal{L}_{\text{TP}}$ solely to the final clean output—which neglects the frequency-dependent dynamics along the diffusion trajectory—we re-projected the prediction back onto the noisy manifold via $\mathcal{Q}$, thereby enforcing consistency at the frequency bands corresponding to each noise level. Specifically, given the predicted clean sample $\hat{x}_0 = f_\theta(x_t, y_0, t)$, the Trajectory Alignment loss is formulated as the cumulative discrepancy between these re-projected estimates states $\hat{x}_t$ and the corresponding target states $x_t$ over the discretized time steps $\mathcal{T}$.

$$\mathcal{L}_{\text{TA}} = \sum_{t \in \mathcal{T}} d_{\text{I}}\big(\mathcal{Q}(\hat{x}_0, y_0, t), \mathcal{Q}(x_0, y_0, t)\big) = \sum_{t \in \mathcal{T}} d_{\text{I}}\left(\hat{x}_t, x_t\right).$$
(10)

Crucially, unlike stochastic sampling, we evaluate this loss over the full discretized time schedule $\mathcal{T}$ (e.g., $|\mathcal{T}| = T = 5$). At each timestep $t \in \mathcal{T}$, the same noise realization is used to generate the paired projections. While identical additive noise would cancel out in linear pixel-wise metrics, our strategy incorporates perceptual losses, where non-linear feature interactions prevent such cancellation. By providing targeted guidance at distinct noise levels, we effectively rectify the tangent vector field of the PF-ODE.

### 3.3. Dual-Reference Structural Rectification

**Analysis of Geometric Decoupling.** While DTM employs perceptual metrics (e.g., LPIPS) to encourage global trajectory alignment, we argue that its objective is constrained by the inherent spatial invariance of deep feature representations. While such metrics effectively mitigate blurriness by matching semantic distributions, they inherently lack sensitivity to precise local geometric variations due to pooling layers and large receptive fields.

To examine this limitation, we conduct an objective–geometry decoupling analysis in Figure. 4, where all quantities are evaluated at an intermediate time step $t'$ sampled during the second training stage. The x-axis measures structural instability, quantified via the Sobel operator $\mathcal{S}(\cdot)$ as Structural MAE $\frac{1}{N}\|\mathcal{S}(f_{\theta'}(\hat{x}_{t'}, y_0, t')) - \mathcal{S}(f_{\theta'}(x_{t'}, y_0, t'))\|_1$, revealing discrepancies in local geometric orientation often overlooked by perceptual objectives. In contrast, the y-axis reports the pixel-wise MAE $\frac{1}{N}\|f_{\theta'}(\hat{x}_{t'}, y_0, t') - f_{\theta'}(x_{t'}, y_0, t')\|_1$, which captures positional alignment.

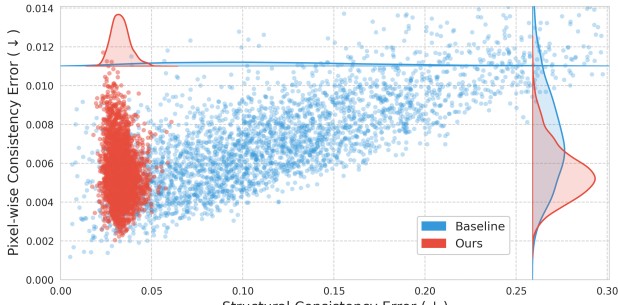

*Figure 4.* Visualization of Geometric Decoupling. Evaluated on 3,000 ImageNet-Test images, the Baseline (blue) achieves positional convergence (low y-axis error) but fails to maintain structural stability (high x-axis variance). In contrast, GTASR (red) minimizes errors on both metrics, effectively resolving the decoupling. Intuitively, accurate pixel-wise alignment should guarantee geometric consistency, as correct pixels naturally manifest correct structures. Surprisingly, our empirical results show that this intuition breaks down under DTM optimization. Specifically, the baseline (blue) exhibits strong compression along the y-axis but severe dispersion along the x-axis, revealing a misalignment between positional convergence and geometric consistency. We refer to this phenomenon as *Geometric Decoupling*. In contrast, GTASR (red) significantly collapses the horizontal variance, demonstrating that our method effectively stabilizes the local geometric structure.

**Structural Error Analysis.** To resolve this decoupling, we do not merely add heuristic constraints; instead, we derive our objectives from a theoretical analysis of the structural error. We define the structural error $e_{\text{tex}}$ as the discrepancy in the gradient domain between the fake and ideal trajectories. Since directly optimizing this global integral is intractable, we seek to control it by bounding its magnitude. By applying the integral triangle inequality, we derive a cumulative upper bound for the error (the detailed definition and derivation are provided in the Appendix B):

$$\|e_{\text{tex}}\|_1 \leq \int_0^T \Big( \underbrace{\|\mathcal{S}(f_{\theta'}(\hat{x}_s, y_0, s)) - \mathcal{S}(f_{\theta'}(x_s, y_0, s))\|_1}_{\text{Term I: Consistency Gap}}$$
$$+ \underbrace{\|\mathcal{S}(f_\theta(x_s, y_0, s)) - \mathcal{S}(x_0)\|_1}_{\text{Term II: Target Bias}} \Big) \cdot |\gamma(s)| \mathrm{d}s.$$
(11)

This derivation implies that the global structural error can be effectively suppressed by tightening this upper bound. Specifically, it reveals two necessary conditions: (1) minimizing the Consistency Gap (Term I) to ensure the trajectory is self-consistent locally, and (2) minimizing the Target Bias (Term II) to anchor the trajectory to the true geometric structure. Drawing inspiration from (Liu et al., 2023; Lee et al., 2024), which demonstrate that theoretical $L_1$ objectives can

*Table 1.* Quantitative comparison on ImageNet-Test and RealSR. The best and second-best results are highlighted in **bold** and underline.

| Datasets | Methods | PSNR↑ | SSIM↑ | LPIPS↓ | CLIPIQA↑ | MUSIQ↑ | MANIQA↑ | NIQE↓ | LIQE↑ | TOPIQ↑ |
|---|---|---|---|---|---|---|---|---|---|---|
| **ImageNet-Test** | BSRGAN | 24.42 | 0.6585 | 0.2585 | 0.5812 | 54.70 | 0.3866 | 6.08 | 3.90 | 0.6078 |
| | Real-ESRGAN | 24.04 | 0.6650 | 0.2540 | 0.5234 | 52.54 | 0.3671 | 6.07 | 3.83 | 0.5545 |
| | StableSR-200 | 20.73 | 0.4650 | 0.3927 | 0.6263 | 56.85 | 0.4272 | 8.32 | 3.49 | 0.5818 |
| | ResShift-15 | 24.95 | 0.6741 | 0.2377 | 0.5823 | 53.11 | 0.4168 | 6.90 | 3.45 | 0.5877 |
| | ResShift-4 | **25.02** | **0.6833** | 0.2074 | 0.5993 | 52.10 | 0.3888 | 7.33 | 3.40 | 0.5679 |
| | SinSR-1 | 24.70 | 0.6635 | 0.2187 | 0.6079 | 53.53 | 0.4152 | 6.29 | 3.58 | 0.5950 |
| | UPSR-5 | 23.76 | 0.6290 | 0.2464 | 0.6149 | 59.20 | 0.4695 | **5.24** | 4.00 | 0.6491 |
| | CTMSR-1 | 24.73 | 0.6660 | 0.1969 | 0.6912 | 60.17 | 0.4857 | 5.66 | 4.08 | 0.6793 |
| | GTASR-1 (Ours) | 24.01 | 0.6520 | **0.1916** | **0.7475** | **65.09** | **0.5826** | 5.83 | **4.47** | **0.7423** |
| **RealSR** | BSRGAN | **26.50** | **0.7746** | 0.2685 | 0.5437 | 63.59 | 0.3702 | 4.65 | 3.38 | 0.5459 |
| | Real-ESRGAN | 25.85 | 0.7734 | 0.2729 | 0.4903 | 59.69 | 0.3675 | 4.68 | 3.37 | 0.5101 |
| | StableSR-200 | 25.78 | 0.7555 | **0.2664** | 0.4125 | 48.35 | 0.3023 | 5.85 | 2.61 | 0.4459 |
| | ResShift-15 | 26.45 | 0.7517 | 0.3620 | 0.6006 | 58.92 | 0.3898 | 6.29 | 3.11 | 0.4939 |
| | ResShift-4 | 25.46 | 0.7257 | 0.3719 | 0.5835 | 56.20 | 0.3480 | 7.34 | 2.80 | 0.4420 |
| | SinSR-1 | 25.97 | 0.7069 | 0.4009 | 0.6667 | 59.23 | 0.4078 | 6.25 | 3.01 | 0.5007 |
| | UPSR-5 | 26.44 | 0.7587 | 0.2872 | 0.5472 | 64.60 | 0.3803 | **4.01** | 3.03 | 0.5638 |
| | CTMSR-1 | 26.19 | 0.7655 | 0.2934 | 0.6450 | 64.71 | 0.4154 | 4.66 | 3.36 | 0.5965 |
| | GTASR-1 (Ours) | 26.34 | 0.7633 | 0.3038 | **0.6931** | **66.50** | **0.4694** | 4.52 | **3.53** | **0.6732** |

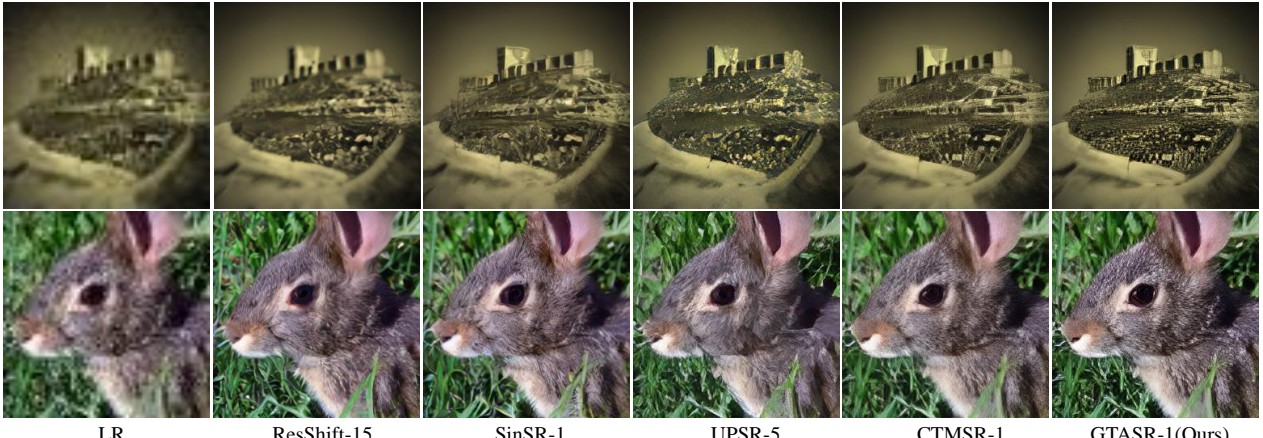

| LR | ResShift-15 | SinSR-1 | UPSR-5 | CTMSR-1 | GTASR-1(Ours) |

*Figure 5.* Visual comparisons of different methods on two synthetic examples of the ImageNet-Test dataset.

be effectively substituted with perceptual metrics to enhance visual quality, we propose Dual-Reference Structural Rectification (DRSR), which explicitly minimizes these components using perceptual constraints (see Appendix C.3).

**(i). Minimizing Consistency Gap via Stability Loss.** To minimize the local inconsistency identified in Term I, we introduce the Stability Loss ($\mathcal{L}_{\text{Stab}}$). Specifically, mirroring the trajectory discrepancy formulation in Eq. 5, we quantify the structural discrepancy $\Delta_{t'}^{\text{Stab}}$ by applying the $\mathcal{S}(\cdot)$ to the endpoint of fake and real trajectories generated by $f_{\theta'}$:

$$\Delta_{t'}^{\text{Stab}} = \mathcal{S}(f_{\theta'}(\hat{x}_{t'}, y_0, t')) - \mathcal{S}(f_{\theta'}(x_{t'}, y_0, t')). \quad (12)$$

Accordingly, the Stability Loss is defined as the expected magnitude of this discrepancy:

$$\mathcal{L}_{\text{Stab}} = \mathbb{E}_{x,t,t'}\Big[\omega(t') \cdot d_{\text{II}}\big(\hat{x}_0, \underbrace{(\hat{x}_0 - \Delta_{t'}^{\text{Stab}})}_{\text{Stop-Gradient}}\big)\Big]. \quad (13)$$

**(ii). Minimizing Target Bias via Rectification Loss.** In contrast, to correct the global deviation characterized in Term II, we employ the Rectification Loss ($\mathcal{L}_{\text{Rect}}$). Specifically, this objective utilizes the real image $x_0$ as a rigorous geometric reference to rectify the orientation of the spatial derivatives extracted via Sobel:

$$\mathcal{L}_{\text{Rect}} = \mathbb{E}_{x,t,t'}\Big[d_{\text{II}}\big((\mathcal{S}(f_{\theta}(x_{t'}, y_0, t')), \mathcal{S}(x_0)\big)\Big]. \quad (14)$$

Together these two losses form a complementary dual-constraint system: neglecting either weakens the control over the structural error characterized in Eq. 11.

*Table 2.* Quantitative results of models on two real-world datasets. The best and second best results are highlighted in **bold** and underline.

| Methods | RealLQ250 | | | | | | RealSet65 | | | | | |
|---|---|---|---|---|---|---|---|---|---|---|---|---|
| | CLIPIQA↑ | MUSIQ↑ | MANIQA ↑ | NIQE↓ | LIQE↑ | TOPIQ↑ | CLIPIQA↑ | MUSIQ↑ | MANIQA ↑ | NIQE↓ | LIQE↑ | TOPIQ↑ |
| BSRGAN | 0.5690 | 63.51 | 0.3514 | 4.54 | 3.31 | 0.5386 | 0.6162 | 65.58 | 0.3888 | 4.58 | 3.72 | 0.5809 |
| Real-ESRGAN | 0.5435 | 62.52 | 0.3565 | 4.13 | 3.34 | 0.5079 | 0.5990 | 63.22 | 0.3893 | 4.41 | 3.60 | 0.5365 |
| StableSR-200 | 0.4001 | 48.83 | 0.2694 | 5.84 | 2.46 | 0.4405 | 0.4492 | 48.75 | 0.3098 | 5.74 | 2.91 | 0.4644 |
| ResShift-15 | 0.6006 | 58.92 | 0.3898 | 6.29 | 3.11 | 0.4939 | 0.6496 | 61.16 | 0.4079 | 6.16 | 3.34 | 0.5397 |
| ResShift-4 | 0.5835 | 56.20 | 0.3480 | 7.34 | 2.80 | 0.4420 | 0.6274 | 59.39 | 0.3621 | 6.74 | 3.16 | 0.4951 |
| SinSR-1 | 0.6667 | 59.23 | 0.4078 | 6.25 | 3.01 | 0.5007 | 0.7199 | 62.61 | 0.4358 | 6.02 | 3.50 | 0.5694 |
| UPSR-5 | 0.5472 | 64.60 | 0.3803 | **4.01** | 3.03 | 0.5638 | 0.6041 | 63.66 | 0.3938 | **4.28** | 3.38 | 0.5647 |
| CTMSR-1 | 0.6703 | 68.02 | 0.4173 | 4.58 | 3.33 | 0.6340 | 0.6883 | 67.28 | 0.4356 | 4.49 | 3.70 | 0.6293 |
| GTASR-1(Ours) | **0.7355** | **70.90** | **0.4870** | 4.32 | **3.79** | **0.7047** | **0.7491** | **69.49** | **0.4989** | 4.45 | **3.97** | **0.6937** |

# 4. Experiments

## 4.1. Experimental settings

**Training datasets.** Following established protocols (Yue et al., 2023; You et al., 2025), we utilize $256 \times 256$ patches randomly cropped from ImageNet (Deng et al., 2009) as HR training data. The corresponding LR images are synthesized from these HR counterparts using the complex degradation pipeline of Real-ESRGAN (Wang et al., 2021).

**Testing dataset.** We evaluate our methods on ImageNet-Test (Deng et al., 2009), a synthetic dataset comprising 3,000 paired images. The LQ-HQ pairs involve a $\times 4$ SR task scaling from $64 \times 64$ to $256 \times 256$. Additionally, we utilize three real-world datasets for validation: RealSR (Cai et al., 2019), RealLQ250 (Ai et al., 2024), and RealSet65 (Yue et al., 2023). RealSR consists of uncropped paired data, whereas RealLQ250 and RealSet65 are datasets without GT, collected from old photos, movie scenes, and social media exhibiting diverse real-world degradations.

**Evaluation metrics.** We evaluate all methods using both full-reference and no-reference image quality metrics. For full-reference assessment, we employ PSNR and SSIM (Wang et al., 2004), computed on the Y channel in YCbCr space to evaluate reconstruction fidelity, alongside LPIPS (Zhang et al., 2018) to measure perceptual similarity. The no-reference metrics include CLIPIQA (Wang et al., 2023a), MUSIQ (Ke et al., 2021), MANIQA (Yang et al., 2022), NIQE (Zhang et al., 2015), LIQE (Zhang et al., 2023), and TOPIQ (Chen et al., 2024), which estimate perceptual quality without reference images.

**Implementation details.** We employ a two-stage training scheme with a fixed learning rate of $5 \times 10^{-5}$ and a total batch size of 32. In Stage I, the model is trained from scratch for 500K iterations using the $\mathcal{L}_{CT}$ and $\mathcal{L}_{TA}$ strategies. In Stage II, we utilize the frozen model from the first stage as a reference and fine-tune for an additional 4K iterations, incorporating $\mathcal{L}_{DTM}$, $\mathcal{L}_{Stab}$, $\mathcal{L}_{Rect}$, and $\mathcal{L}_{CT}$. For more specific details, please refer to the Appendix C.

*Table 3.* Computational efficiency and performance comparison. We report Runtime for $128 \times 128$ input on a single RTX 4090 GPU and present perceptual metrics evaluated on ImageNet-Test.

| Methods | Runtime (s)↓ | LPIPS↓ | MANIQA↑ | CLIPIQA↑ | TOPIQ↑ |
|---|---|---|---|---|---|
| StableSR-200 | 11.21 | 0.3927 | 0.4272 | 0.6263 | 0.5818 |
| ResShift-15 | 0.93 | 0.2377 | 0.4168 | 0.5823 | 0.5877 |
| UPSR-5 | 0.35 | 0.2464 | 0.4695 | 0.6149 | 0.6491 |
| SinSR-1 | 0.12 | 0.2187 | 0.4152 | 0.6079 | 0.5950 |
| CTMSR-1 | **0.08** | 0.1969 | 0.4857 | 0.6912 | 0.6793 |
| GTASR-1 (Ours) | **0.08** | **0.1916** | **0.5826** | **0.7475** | **0.7423** |

*Table 4.* Ablation study of the Stage-I training objectives on ImageNet-Test.

| Method | PSNR↑ | LPIPS↓ | CLIPIQA↑ | MUSIQ↑ | MANIQA↑ | LIQE↑ | TOPIQ↑ |
|---|---|---|---|---|---|---|---|
| w/o $\mathcal{L}_{TA}$ | 24.65 | 0.2073 | 0.6060 | 55.38 | 0.3830 | 3.86 | 0.5939 |
| w/ $\mathcal{L}_{TA}$ | **25.05** | **0.1856** | **0.6291** | **58.40** | **0.4457** | **4.05** | **0.6433** |

*Table 5.* Ablation study of the Stage-II training objectives on ImageNet-Test.

| $\mathcal{L}_{DTM}$ | $\mathcal{L}_{Stab}$ | $\mathcal{L}_{Rect}$ | PSNR↑ | LPIPS↓ | CLIPIQA↑ | MUSIQ↑ | MANIQA↑ | LIQE↑ | TOPIQ↑ |
|---|---|---|---|---|---|---|---|---|---|
| ✓ | | | **24.76** | 0.2019 | 0.6965 | 58.91 | 0.5070 | 3.77 | 0.6863 |
| ✓ | ✓ | | 24.00 | 0.1929 | 0.7351 | 63.43 | 0.5676 | 4.42 | 0.7349 |
| ✓ | | ✓ | 24.62 | 0.2000 | 0.7040 | 60.41 | 0.5403 | 3.89 | 0.6953 |
| ✓ | ✓ | ✓ | 24.01 | **0.1916** | **0.7475** | **65.09** | **0.5826** | **4.47** | **0.7423** |

## 4.2. Comparison with the state of the art

To demonstrate the superiority of our method, we compare it against representative SR approaches, including GAN-based methods (BSRGAN (Zhang et al., 2021), Real-ESRGAN (Wang et al., 2021)), multi-step diffusion-based methods (StableSR (Wang et al., 2024a), ResShift (Yue et al., 2023), UPSR(Zhang et al., 2025)), and one-step diffusion-based methods (SinSR (Wang et al., 2024b), CTMSR (You et al., 2025)). Comparisons with advanced methods based on T2I model distillation are presented in the Appendix D.2.

**Qualitative Comparisons.** Figures 5 and 6 present visual comparisons with representative baselines. Our method shows a clear advantage in recovering intricate high-frequency details. As illustrated in Figure 5, it effectively recovers fine architectural contours, producing results that are noticeably clearer and more structurally well-defined. Similarly, for animal samples, our method faithfully reconstructs realistic fur and skin details. These results highlight the superiority of our framework in achieving both structural integrity and perceptual realism.

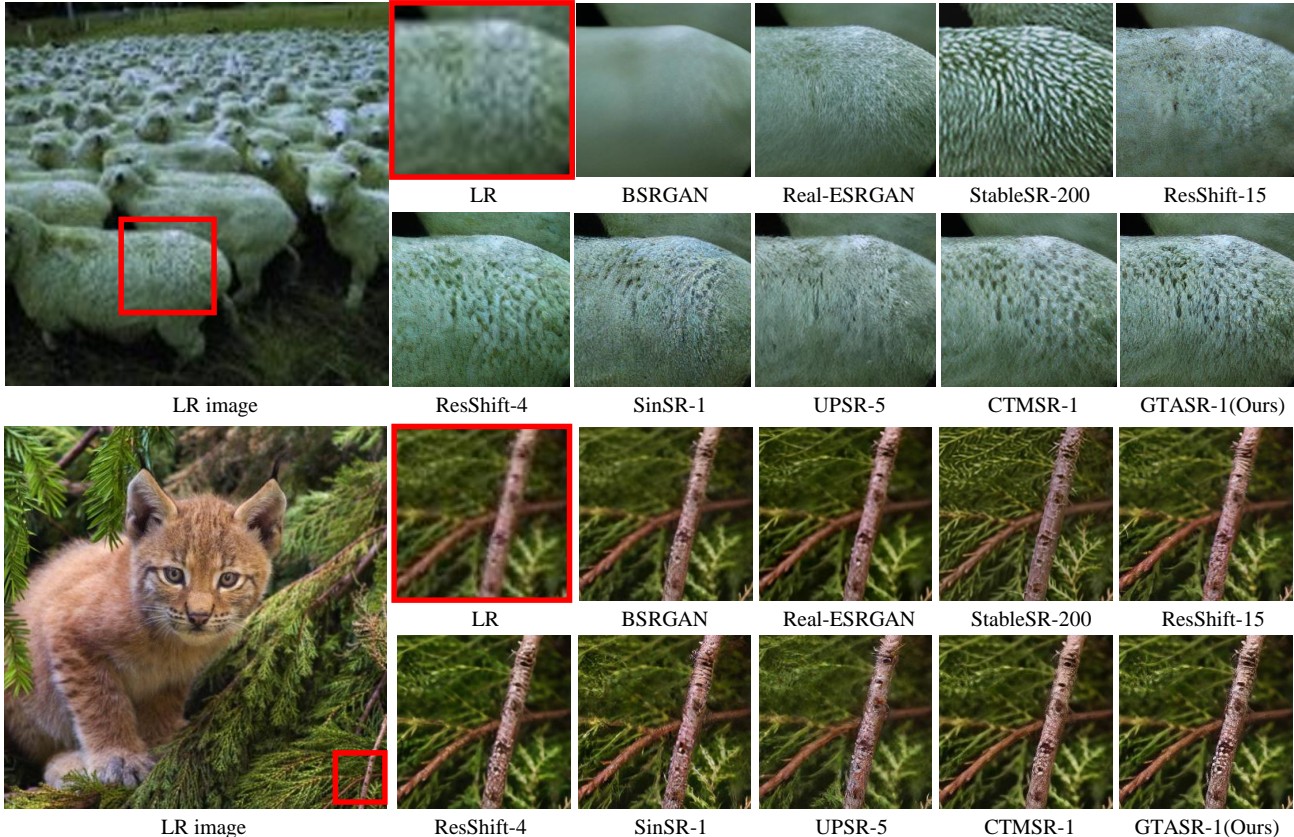

*Figure 6.* Visual comparisons of different methods on two examples of real-world datasets. Please zoom in for more details.

**Quantitative Comparisons.** As shown in Table 1, GTASR achieves lower PSNR/SSIM scores; however, richer texture generation is often accompanied by such reductions (Yue et al., 2025; Lin et al., 2025). When evaluated with perceptual quality metrics, GTASR exhibits a clear advantage. On the synthetic ImageNet-Test, GTASR achieves a MANIQA score of 0.5826, significantly outperforming CTMSR's score of 0.4857. This advantage is further amplified on the real-world RealLQ250 dataset in Table 2, where GTASR surpasses CTMSR in TOPIQ by a large margin, scoring 0.7047 compared to 0.6340.

**Evaluation of efficiency.** We evaluate inference efficiency and perceptual quality by comparing GTASR with representative diffusion-based methods. As shown in Table 3, benefiting from one-step inference, GTASR reduces inference latency to only 8.6% of ResShift-15 and 0.7% of StableSR-200, while remaining faster than other one-step methods SinSR. Despite its high efficiency, GTASR consistently delivers superior perceptual quality, achieving the best performance across all reported perceptual metrics. Additional efficiency comparisons are provided in the Appendix D.1.

## 5. Ablation Study

**Effectiveness of Trajectory Alignment**. As shown in Table 4, integrating $\mathcal{L}_{\text{TA}}$ improves reconstruction fidelity by a notable margin, with an increase of 0.40 dB in PSNR and a reduction of 0.0217 in LPIPS. We attribute these improvements to the capability of $\mathcal{L}_{\text{TA}}$ in rectifying the tangent vector field, which effectively mitigates consistency drift.

**Dual-Reference Structural Rectification.** Table 5 illustrates the synergistic impact of our proposed loss terms. Specifically, the introduction of $\mathcal{L}_{\text{Stab}}$ significantly enhances perceptual quality, boosting the MANIQA score from 0.5070 to 0.5676 and improving TOPIQ from 0.6863 to 0.7349, which we attribute to its ability to suppress structural inconsistency. Similarly, $\mathcal{L}_{\text{Rect}}$ improves performance by correcting bias in the model's structural predictions. Except for PSNR, our proposed method consistently outperforms all baselines across the remaining 6 evaluation metrics. These results confirm that $\mathcal{L}_{\text{Stab}}$ and $\mathcal{L}_{\text{Rect}}$ are highly complementary: while $\mathcal{L}_{\text{Stab}}$ ensures geometric stability, $\mathcal{L}_{\text{Rect}}$ refines texture fidelity. Due to space limitations, more ablation studies are provided in the Appendix D.

## 6. Conclusion

In this paper, we propose GTASR (Geometric Trajectory Alignment Super-Resolution), an efficient method that addresses the critical limitations of consistency drift and geometric decoupling to enable high-realism one-step generation. We first introduce a Trajectory Alignment (TA) strategy to rectify the tangent vector field via full-path projection, thereby mitigating the error accumulation inherent in the generative process. To further bridge the geometric gap, we propose Dual-Reference Structural Rectification (DRSR), which enforces structural constraints to effectively recover intricate high-frequency details. Extensive experimental results demonstrate that our method delivers performance competitive with the state-of-the-art baselines while maintaining minimal inference latency.

## Acknowledgements

This work was supported by the National Natural Science Foundation of China (Grant Nos. 62372082 and 62572234), the Fundamental Research Funds for the Central Universities(No. ZYGX2024Z017), and Shenzhen Natural Science Foundation (No. JCYJ20240813114206010).

## Impact Statement

This paper presents work whose goal is to advance the field of Machine Learning. There are many potential societal consequences of our work, none which we feel must be specifically highlighted here.

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

# Appendix

The appendix is organized as follows: Appendix A and Appendix B provide theoretical derivations for the PF-ODE and structural error analysis. Appendix C details implementation settings and the full training algorithm. Finally, Appendix D presents additional efficiency benchmarks and visual comparisons.

## A. Derivation of the Probability Flow ODE

This section details the derivation of the Probability Flow ODE (PF-ODE) under the ResShift noise schedule. By analyzing the forward diffusion dynamics, we rigorously deduce the deterministic differential equation governing the generative trajectory, leading to Eq. 8 in the main text.

### A.1. Forward Process Dynamics

Recall the specific forward diffusion process defined in Eq. 1:

$$x_t = \mathcal{Q}(x_0, y_0, t) = x_0 + \alpha_t e_0 + \sigma_t \epsilon, \tag{15}$$

where $e_0 = y_0 - x_0$ represents the residual, and $\epsilon \sim \mathcal{N}(0, I)$ is the standard Gaussian noise. To construct the Ordinary Differential Equation (ODE) that describes the continuous dynamics of this process, we first determine the instantaneous rate of change of the state $x_t$. Differentiating Eq. 15 with respect to time $t$, we obtain the time-dependent evolution velocity:

$$\frac{\mathrm{d}x_t}{\mathrm{d}t} = \dot{\alpha}_t e_0 + \dot{\sigma}_t \epsilon. \tag{16}$$

However, this equation depends on the unknown initial residual $e_0$ and noise $\epsilon$. To derive the corresponding Probability Flow ODE, we replace the random noise $\epsilon$ and the fixed residual $e_0$ with their instantaneous estimates provided by the model. Let $f_\theta(x_t, y_0, t)$ denote the model's prediction of the high-resolution image $x_0$. Consequently, the estimated residual is $e_\theta = y_0 - f_\theta(x_t, y_0, t)$. By rearranging Eq. 15, the implicit noise term $\epsilon$ can be expressed as:

$$\epsilon = \frac{x_t - x_0 - \alpha_t e_0}{\sigma_t} \implies \epsilon_\theta = \frac{x_t - f_\theta(x_t, y_0, t) - \alpha_t(y_0 - f_\theta(x_t, y_0, t))}{\sigma_t}. \tag{17}$$

### A.2. Derivation of the PF-ODE

Substituting these estimates into the general form of the PF-ODE (Eq. 3), we obtain the following:

$$\mathrm{d}x_t = \left[ \dot{\alpha}_t(y_0 - f_\theta(x_t, y_0, t)) + \dot{\sigma}_t \left( \frac{x_t - f_\theta(x_t, y_0, t) - \alpha_t(y_0 - f_\theta(x_t, y_0, t))}{\sigma_t} \right) \right] \mathrm{d}t. \tag{18}$$

Grouping the terms associated with the residual prior $(y_0 - f_\theta(x_t, y_0, t))$ and the state term $(x_t - f_\theta(x_t, y_0, t))$ yields:

$$\mathrm{d}x_t = \left[ \left( \dot{\alpha}_t - \frac{\dot{\sigma}_t \alpha_t}{\sigma_t} \right) (y_0 - f_\theta(x_t, y_0, t)) + \frac{\dot{\sigma}_t}{\sigma_t} (x_t - f_\theta(x_t, y_0, t)) \right] \mathrm{d}t. \tag{19}$$

In our implementation, we adopt a specific noise schedule defined as a power function of the normalized time step:

$$\alpha_t = \sigma_t = \left( \frac{t}{T} \right)^n, \tag{20}$$

where $n$ is a constant scaling factor. To verify the cancellation of the residual term, we first compute the time derivatives:

$$\dot{\alpha}_t = \dot{\sigma}_t = \frac{d}{dt} \left[ \left( \frac{t}{T} \right)^n \right] = \frac{n}{T} \left( \frac{t}{T} \right)^{n-1} = \frac{n}{t} \cdot \left( \frac{t}{T} \right)^n = \frac{n}{t} \alpha_t = \frac{n}{t} \sigma_t. \tag{21}$$

Substituting these explicit forms into the coefficient of the residual term $(y_0 - f_\theta(x_t, y_0, t))$, we obtain:

$$\dot{\alpha}_t - \frac{\dot{\sigma}_t \alpha_t}{\sigma_t} = \left( \frac{n}{t} \alpha_t \right) - \frac{\left( \frac{n}{t} \sigma_t \right) \alpha_t}{\sigma_t} = \frac{n}{t} \alpha_t - \frac{n}{t} \alpha_t = 0. \tag{22}$$

This rigorous substitution confirms that under the schedule $\alpha_t = \sigma_t \propto t^n$, the drift term associated with the residual prior $y_0$ is mathematically eliminated. The differential equation thus simplifies to:

$$\mathrm{d}x_t = \frac{\dot{\sigma}_t}{\sigma_t}(x_t - f_\theta(x_t, y_0, t))\mathrm{d}t = \frac{n}{t}(x_t - f_\theta(x_t, y_0, t))\mathrm{d}t. \tag{23}$$

Integrating this simplified differential equation from the current time step $t$ back to $t = 0$ allows us to reconstruct the complete trajectory. By substituting the derived differential form (Eq. 9) into the integral path, we obtain the explicit formulation for recovering the clean data $x_0$:

$$x_0 \approx x_t + \int_t^0 \underbrace{\frac{n}{s}\big(x_s - f_\theta(x_s, y_0, s)\big)}_{\text{Derived Drift Dynamics}} \mathrm{d}s. \tag{24}$$

This equation reveals that the reconstruction process is governed by accumulating the instantaneous restoration directions $(x_s - f_\theta(x_s, y_0, s))$, scaled by the time-dependent factor $\frac{n}{s}$ specific to our noise schedule.

## B. Sobel-Based Structural Error Analysis and Optimization Motivation

This section provides a structural error analysis to motivate the design of Dual-Reference Structural Rectification (DRSR). Since directly optimizing the terminal endpoint requires backpropagating through the discretized solver—rendering the process computational intractable and numerically unstable—we instead analyze how structural discrepancies accumulate along the generation trajectory. This derivation establishes a tractable upper bound that serves as an effective optimization surrogate, explaining why constraining intermediate structural behavior is essential for enhancing both stability and perceptual fidelity.

### B.1. Integral Analysis in a Structure-Aware Domain

To characterize the evolution of fine structural details, we extend the trajectory analysis from the pixel domain to a structure-aware domain. We employ the Sobel operator, denoted as $\mathcal{S}(\cdot)$, as a fixed, linear, and time-independent operator that highlights local structural variations. Although $\mathcal{S}(\cdot)$ is a discrete approximation, its linearity allows it to commute with temporal integration. Applying $\mathcal{S}(\cdot)$ to both sides of Eq. 24, we obtain the following Sobel-based integral formulation:

$$\mathcal{S}(x_0) \approx \mathcal{S}(x_t) + \int_t^0 \frac{n}{s}\Big(\mathcal{S}(x_s) - \mathcal{S}\big(f_\theta(x_s, y_0, s)\big)\Big)\mathrm{d}s. \tag{25}$$

This formulation describes how structural features evolve along the PF-ODE based on the model-predicted dynamics.

### B.2. Analysis of Distinct Structural Trajectories

In line with prior trajectory-based analyses (You et al., 2025), based on Eq. 25, we examine three structurally distinct trajectories to identify the sources of structural degradation under learned dynamics.

- **Fake Trajectory.** This trajectory represents the actual generation path induced by the target model $f_{\theta'}$. The intermediate state $\hat{x}_s$ is obtained via a re-noising process, defined as $\hat{x}_s = \mathcal{Q}\big(f_\theta(x_t, y_0, t), y_0, s\big)$. Its structural reconstruction is given as follows:

$$\mathcal{S}(x_0^{\text{fake}}) = \mathcal{S}(\hat{x}_t) + \int_t^0 \frac{n}{s}\left(\mathcal{S}(\hat{x}_s) - \mathcal{S}\big(f_{\theta'}(\hat{x}_s, y_0, s)\big)\right)\mathrm{d}s. \tag{26}$$

- **Real Trajectory.** This trajectory is produced by the target model $f_{\theta'}$ conditioned on the standard forward-diffused states $x_s = \mathcal{Q}\big(x_0, y_0, s\big)$. It serves as a practical reference trajectory, but it deviate from the ideal trajectory due to the model's limited capacity:

$$\mathcal{S}(x_0^{\text{real}}) = \mathcal{S}(x_t) + \int_t^0 \frac{n}{s}\left(\mathcal{S}(x_s) - \mathcal{S}\big(f_{\theta'}(x_s, y_0, s)\big)\right)\mathrm{d}s. \tag{27}$$

- **Ideal Trajectory.** This trajectory represents the idealized structural mapping, where the target structure is preserved at all timesteps. It serves as a conceptual reference, satisfying $f^*(x_s, y_0, s) \equiv x_0$:

$$\mathcal{S}(x_0) = \mathcal{S}(x_t) + \int_t^0 \frac{n}{s} \left(\mathcal{S}(x_s) - \mathcal{S}(x_0)\right) ds. \tag{28}$$

### B.3. Trajectory-Based Decomposition of Structural Error

We analyze the structural error from a trajectory stability perspective. Instead of establishing a continuous-time optimality result, this subsection aims to characterize how intermediate structural deviations are propagated and accumulated along the discretized PF-ODE trajectory, thereby motivating trajectory-level rectification. We define the structural (texture) error at the endpoint $t = 0$ as follows:

$$e_{\text{tex}} = \mathcal{S}(x_0^{\text{fake}}) - \mathcal{S}(x_0), \tag{29}$$

We concentrate our analysis on the fake trajectory and ideal trajectory because it represents the evolvable path determined by the learnable parameters $\theta$, rendering the structural deviations in $\hat{x}_s$ parameter-dependent and rectifiable. In contrast, the real trajectory acts merely as a static geometric guide generated by a frozen model; any deviation it exhibits from $x_0$ represents an irreducible bias that remains constant and cannot be mitigated by updating $\theta$.

Starting from the initial boundary at $t = T$, according to Eq. 26 Eq. 28 and Eq. 29 the endpoint structural error can be expressed as the accumulated effect of trajectory discrepancies along the PF-ODE evolution:

$$e_{\text{tex}} = \underbrace{\left(\mathcal{S}(\hat{x}_T) - \mathcal{S}(x_T)\right)}_{g_T} + \int_T^0 \frac{n}{s} \left[ \underbrace{\left(\mathcal{S}(\hat{x}_s) - \mathcal{S}(x_s)\right)}_{g_s} - \underbrace{\left(\mathcal{S}\big(f_{\theta'}(\hat{x}_s, y_0, s)\big) - \mathcal{S}(x_0)\right)}_{\mathcal{D}(\hat{x}_s, y_0, s)} \right] ds. \tag{30}$$

**Analysis of Boundary Term ($g_T$).** The term $g_T$ accounts for potential mismatch at initialization. In practice, the generative trajectory is initialized from the exact same forward-diffused state as the reference trajectory. Specifically, given $x_T = x_0 + \alpha_T e_0 + \sigma_T \epsilon = y_0 + \epsilon$, we enforce $\hat{x}_T \equiv x_T$, which yields:

$$g_T = \mathcal{S}(\hat{x}_T) - \mathcal{S}(x_T) = 0. \tag{31}$$

**Dynamics of the Structural Discrepancy ($g_s$).** We define the instantaneous structural discrepancy at time $s$ as:

$$g_s = \mathcal{S}(\hat{x}_s) - \mathcal{S}(x_s). \tag{32}$$

Although $\mathcal{S}(\cdot)$ does not correspond to a spatial gradient, it is a fixed, linear, and time-independent operator, satisfying $\mathcal{S}(ax_1 + bx_2) = a\mathcal{S}(x_1) + b\mathcal{S}(x_2)$ and $\partial_s \mathcal{S}(x) = \mathcal{S}(\partial_s x)$. Taking the time derivative of $g_s$ yields:

$$\begin{aligned}
\frac{dg_s}{ds} &= \frac{d}{ds}\left(\mathcal{S}(\hat{x}_s) - \mathcal{S}(x_s)\right) && \text{(time differentiation)} \\
&= \mathcal{S}\left(\frac{d\hat{x}_s}{ds} - \frac{dx_s}{ds}\right) && \text{(linearity and time-independence of } \mathcal{S}(\cdot)) \\
&= \mathcal{S}\left(\frac{n}{s}\big(\hat{x}_s - f_{\theta'}(\hat{x}_s, y_0, s)\big) - \frac{n}{s}\big(x_s - x_0\big)\right) && \text{(by PF-ODE, Eq. 24)} \\
&= \frac{n}{s}\left[\mathcal{S}(\hat{x}_s - x_s) - \mathcal{S}\big(f_{\theta'}(\hat{x}_s, y_0, s) - x_0\big)\right] && \text{(rearranging terms)} \\
&= \frac{n}{s}\big(g_s - \mathcal{D}(\hat{x}_s, y_0, s)\big),
\end{aligned} \tag{33}$$

where $\mathcal{D}(\hat{x}_s, y_0, s)$ denotes the structure-aware driver term. This formulation reveals that intermediate structural discrepancies are propagated in a linear and non-self-correcting manner along the PF-ODE trajectory. As a result, deviations introduced at intermediate steps may persist and contribute cumulatively to the endpoint error. Accordingly, $g_s$ can be expressed as the response of a linear operator to the driver term $\mathcal{D}$:

$$g_s = \mathcal{K}(s)\big[\mathcal{D}(\hat{x}_\tau, y_0, \tau)\big], \tag{34}$$

where $\mathcal{K}(s)$ characterizes the propagation of structural discrepancies. Overall, Eq. (30) reveals that endpoint structural fidelity is intrinsically bounded by the stability of the intermediate trajectory. While continuous-time theory guarantees a unique solution, in practice, trajectory-level regularization serves as a crucial countermeasure against the numerical instability and error accumulation inherent in discretized neural ODE solvers.

### B.4. Error Accumulation via Upper-Bound Relaxation

By substituting Eq. 31 and Eq. 34 into Eq. 30, we obtain:

$$e_{\text{tex}} = \int_0^T \gamma(s)\,\mathcal{D}(\hat{x}_s, y_0, s)\,\mathrm{d}s, \tag{35}$$

where $\gamma(s)$ is an effective weighting function induced by the discretized dynamics. Although cancellation may occur in principle, relying on such behavior provides no guarantee of stable optimization. We therefore derive a sign-invariant surrogate objective via the integral triangle inequality:

$$\|e_{\text{tex}}\|_1 \leq \int_0^T |\gamma(s)| \cdot \|\mathcal{D}(\hat{x}_s, y_0, s)\|_1\,\mathrm{d}s. \tag{36}$$

Importantly, this bound does not establish equivalence between minimizing trajectory-level discrepancies and minimizing the final error. Instead, it provides a relaxed objective that enables dense structural supervision under learned and discretized dynamics. Applying the triangle inequality yields:

$$\begin{aligned}
\|\mathcal{D}(\hat{x}_s, y_0, s)\|_1 &= \left\|\mathcal{S}\big(f_{\theta'}(\hat{x}_s, y_0, s)\big) - \mathcal{S}(x_0)\right\|_1 \\
&\leq \underbrace{\left\|\mathcal{S}\big(f_{\theta'}(\hat{x}_s, y_0, s)\big) - \mathcal{S}\big(f_{\theta'}(x_s, y_0, s)\big)\right\|_1}_{\text{Term I: Consistency Gap}} + \underbrace{\left\|\mathcal{S}\big(f_{\theta'}(x_s, y_0, s)\big) - \mathcal{S}(x_0)\right\|_1}_{\text{Term II: Target Bias}}.
\end{aligned} \tag{37}$$

This inequality demonstrates that minimizing the total error requires minimizing its upper bound, consisting of two distinct sources. However, directly minimizing Term II is infeasible for two reasons: first, the reference trajectory $x_s$ is fixed, and second, the target parameters $\theta'$ are updated via a stop-gradient mechanism, preventing gradient flow for updates. Nevertheless, the periodic synchronization ($\theta' \leftarrow \theta$) ensures parameter alignment, implying functional proximity (i.e., $f_\theta(x_s, y_0, s) \approx f_{\theta'}(x_s, y_0, s)$). Consequently, Term II can be reformulated using the online model as $\|\mathcal{S}(f_\theta(x_s, y_0, s)) - \mathcal{S}(x_0)\|_1$. This approximation enables us to treat the online reconstruction error as a learnable proxy to effectively minimize the target bias.

### B.5. Dual-Reference Structural Rectification

The above analysis motivates controlling the final structural error by jointly suppressing the two upper-bound components. The first term captures structural inconsistency between trajectories, while the second reflects deviation from the ground-truth structure. These factors correspond to our proposed objectives: the Stability Loss $\mathcal{L}_{\text{Stab}}$ mitigates trajectory-level structural inconsistency, and the Rectification Loss $\mathcal{L}_{\text{Rect}}$ anchors the prediction to the target structure. Consequently, this dual-reference formulation constitutes a mathematically grounded surrogate for the intractable endpoint optimization, ensuring both numerical stability and perceptual fidelity, as validated by the ablation results in Table 5.

## C. Implementation Details

### C.1. Noise Schedule

Following ResShift (Yue et al., 2023), we unify the drift and diffusion coefficients as a power function of the normalized timestep:

$$\alpha_t = \sigma_t = \left(\frac{t}{T}\right)^n, \tag{38}$$

where $n$ controls the noise growth rate. To stabilize optimization, we employ a convex schedule with $n = 2.5$ in Stage I to prioritize low-uncertainty regions, and transition to a linear schedule of $n = 1$ in Stage II for uniform refinement.

## C.2. Training Strategy and Step Schedule

Following (You et al., 2025), the training proceeds in two distinct stages. In Stage I, the model is trained from scratch for $N_I = 500k$ iterations. In Stage II, we initialize the model with the pre-trained weights from Stage I and fine-tune for an additional 4k iterations. During this second stage, to stabilize optimization, we perform a hard update on the target network $f_{\theta'}$ by directly copying parameters from the online network $f_\theta$ every $N_{II} = 1000$ iterations. We adopt a dynamic step schedule where the total discretization steps are set to $T = 5$ for the first half of training and reduced to $T = 4$ for the remaining half. All experiments are trained on 4 NVIDIA GeForce RTX 4090 GPUs.

## C.3. Metric Function and Weighting Schedule.

**Metric Function.** We employ two stage-specific distance operators, $d_I(\cdot, \cdot)$ and $d_{II}(\cdot, \cdot)$, both defined as weighted combinations of the Learned Perceptual Image Patch Similarity (LPIPS) and the Charbonnier distance:

$$d(x, y) = \lambda_1 \, \text{LPIPS}(x, y) + \lambda_2 \, \text{Charbonnier}(x, y). \tag{39}$$

$d_I(\cdot, \cdot)$ is used in Stage I with $(\lambda_1, \lambda_2) = (0.5, 0.5)$ to jointly account for perceptual and pixel-wise consistency. $d_{II}(\cdot, \cdot)$ is adopted in Stage II with $(\lambda_1, \lambda_2) = (1.0, 0.0)$, reducing the metric to a purely perceptual distance.

**Definition of the Weighting Schedule.** Following (You et al., 2025), the time-dependent weighting term $\omega(t')$ utilized in Eq. 6 and Eq. 13 is defined as follows:

$$\omega(t') = \frac{CS}{|\hat{\boldsymbol{x}}_0 - \boldsymbol{x}_0|_1}, \tag{40}$$

where $S$ and $C$ denote the number of spatial locations and the number of channels, respectively. This weighting factor is designed to normalize the loss magnitude relative to the prediction error at each time step, ensuring balanced gradient contributions throughout the distillation process.

## C.4. Overall Training Objective

The training of GTASR proceeds in two distinct stages. In stage I, we train the model from scratch using Consistency Training ($\mathcal{L}_{CT}$) combined with Trajectory Alignment ($\mathcal{L}_{TA}$) to mitigate consistency drift. The objective is formulated as:

$$\mathcal{L}_{\text{Stage-I}} = \mathcal{L}_{CT} + \lambda_{TA} \mathcal{L}_{TA}, \tag{41}$$

where we empirically set $\lambda_{CT} = 1.0$ to prioritize consistency learning, and $\lambda_{TA} = 0.5$ to provide auxiliary trajectory guidance. In Stage II, we initialize the model with the pre-trained weights from Stage I. To further enhance structural fidelity, we incorporate Distribution Trajectory Matching ($\mathcal{L}_{DTM}$) alongside our proposed Dual-Reference Structural Rectification mechanism ($\mathcal{L}_{Stab}$ and $\mathcal{L}_{Rect}$). The total training objective is defined as:

$$\mathcal{L}_{\text{Stage-II}} = \mathcal{L}_{CT} + \lambda_{DTM} \mathcal{L}_{DTM} + \lambda_{Stab} \mathcal{L}_{Stab} + \lambda_{Rect} \mathcal{L}_{Rect}, \tag{42}$$

where we assign the balancing coefficients as $\lambda_{DTM} = 1.6$, $\lambda_{Stab} = 0.2\lambda_{DTM} = 0.032$, and $\lambda_{Rect} = 1.0$. The overall training process is summarized in Algorithm 1.

# D. Additional Experimental Results

## D.1. Efficiency comparison on GPU

We assess the practical deployability of GTASR by benchmarking its computational efficiency and perceptual performance against state-of-the-art diffusion-based SR methods on a single NVIDIA GeForce RTX 4090 GPU, using inputs of size $128 \times 128$ and producing $512 \times 512$ outputs.

As shown in Table 6, conventional diffusion models such as StableSR incur prohibitive computational costs, requiring 11.21 seconds per inference with a model size of 1928 M parameters. In contrast, GTASR operates in real time, achieving a runtime of only 0.08 seconds with a compact model size of 172 M parameters—matching the fastest baseline (CTMSR) while being significantly more efficient than ResShift-4. More importantly, under this highly constrained computational budget, GTASR consistently delivers the best perceptual quality across all evaluation metrics. For example, GTASR improves the MANIQA score from 0.4857 to 0.5826 compared to CTMSR, while maintaining identical inference latency and parameter count.

---

**Algorithm 1** GTASR Training

---

**Require:** Paired training dataset $(X, Y)$; forward projection operator $\mathcal{Q}(x_0, y_0, t)$; online model $f_\theta$; reference model $f_{\theta^-}$; target model $f_{\theta'}$; stop-gradient operator $\text{sg}(\cdot)$; Sobel operator $\mathcal{S}(\cdot)$; weighting function $\omega(t)$; Stage-I iterations $N_{\text{I}}$; Stage-II iterations $N_{\text{II}}$; target synchronization period $N$; diffusion steps $T$;

**Stage I: Consistency Training with Trajectory Alignment**

1: Initialize $\theta$ randomly
2: **for** $i \leftarrow 1$ **to** $N_{\text{I}}$ **do**
3:  Sample $(x_0, y_0) \sim (X, Y)$
4:  Sample $t \sim \mathcal{U}(1, T)$
5:  Obtain $x_t = \mathcal{Q}(x_0, y_0, t)$ and $x_{t-1} = \mathcal{Q}(x_0, y_0, t-1)$
6:  Predict $\hat{x}_0 \leftarrow f_\theta(x_t, y_0, t)$
7:  Update target parameters $\theta^- \leftarrow \text{sg}(\theta)$
8:  Compute $\mathcal{L}_{\text{CT}} = d_{\text{I}}(f_\theta(x_t, y_0, t), f_{\theta^-}(x_{t-1}, y_0, t-1))$
9:  Sample timestep set $\mathcal{T}$
10:  Compute $\mathcal{L}_{\text{TA}} = \sum_{s \in \mathcal{T}} d_{\text{I}}(\mathcal{Q}(\hat{x}_0, y_0, t), \mathcal{Q}(x_0, y_0, t))$
11:  Compute $\mathcal{L}_{\text{Stage-I}} = \mathcal{L}_{\text{CT}} + \lambda_{\text{TA}} \mathcal{L}_{\text{TA}}$
12:  Update $\theta$ using $\nabla_\theta \mathcal{L}_{\text{Stage-I}}$
13:  $i \leftarrow i + 1$
14: **end for**

**Stage II: Distribution Trajectory Matching with Dual-Reference Structural Rectification**

15: Initialize $\theta' \leftarrow \theta$ (trained in Stage I)
16: **for** $i \leftarrow 1$ **to** $N_{\text{II}}$ **do**
17:  **if** $i \equiv 1 \pmod{N}$ **then**
18:      $\theta' \leftarrow \text{sg}(\theta)$
19:  **end if**
20:  Sample $(x_0, y_0) \sim (X, Y)$
21:  Sample $t \sim \mathcal{U}(1, T)$
22:  Obtain $x_t = \mathcal{Q}(x_0, y_0, t)$ and $x_{t-1} = \mathcal{Q}(x_0, y_0, t-1)$
23:  Predict $\hat{x}_0 \leftarrow f_\theta(x_t, y_0, t)$
24:  Compute $\mathcal{L}_{\text{CT}} = d_{\text{I}}(f_\theta(x_t, y_0, t), f_{\theta^-}(x_{t-1}, y_0, t-1))$
25:  Sample $t' \sim \mathcal{U}(T_{\min}, T_{\max})$
26:  Obtain $x_{t'} = \mathcal{Q}(x_0, y_0, t')$ and $\hat{x}_{t'} = \mathcal{Q}(\hat{x}_0, y_0, t')$
27:  Define $\Delta_{t'}^{\text{DTM}} = f_{\theta'}(\hat{x}_{t'}, y_0, t') - f_{\theta'}(x_{t'}, y_0, t')$
28:  Compute $\mathcal{L}_{\text{DTM}} = \mathbb{E}_{x,t,t'}\left[\omega(t') \cdot d_{\text{II}}(\hat{x}_0, \text{sg}(\hat{x}_0 - \Delta_{t'}^{\text{DTM}}))\right]$
29:  Define $\Delta_{t'}^{\text{Stab}} = \mathcal{S}(f_{\theta'}(\hat{x}_{t'}, y_0, t')) - \mathcal{S}(f_{\theta'}(x_{t'}, y_0, t'))$
30:  Compute $\mathcal{L}_{\text{Stab}} = \mathbb{E}_{x,t,t'}\left[\omega(t') \cdot d_{\text{II}}(\hat{x}_0, \text{sg}(\hat{x}_0 - \Delta_{t'}^{\text{Stab}}))\right]$
31:  Compute $\mathcal{L}_{\text{Rect}} = \mathbb{E}_{x,t,t'}[d_{\text{II}}(\mathcal{S}(f_\theta(x_{t'}, y_0, t')), \mathcal{S}(x_0))]$
32:  Compute $\mathcal{L}_{\text{Stage-II}} = \mathcal{L}_{\text{CT}} + \lambda_{\text{DTM}} \mathcal{L}_{\text{DTM}} + \lambda_{\text{Stab}} \mathcal{L}_{\text{Stab}} + \lambda_{\text{Rect}} \mathcal{L}_{\text{Rect}}$
33:  Update $\theta$ using $\nabla_\theta \mathcal{L}_{\text{Stage-II}}$
34:  $i \leftarrow i + 1$
35: **end for**
36: **Return** converged GTASR $f_\theta$.

---

These results demonstrate that GTASR achieves superior perceptual performance without increasing model size or runtime, effectively breaking the long-standing efficiency–quality trade-off in diffusion-based super-resolution and making it well suited for real-time deployment.

*Table 6.* Computational efficiency and performance comparison. We report Runtime and Parameters (Params) measured with an input size of $128 \times 128$ on a single RTX 4090 GPU. We also present perceptual metrics evaluated on *ImageNet-Test*. **Bold** and underlined indicate the best and second-best performance, respectively.

| Methods | Runtime (s)↓ | Params (M)↓ | LPIPS↓ | MUSIQ↑ | MANIQA↑ | CLIPIQA↑ | LIQE↑ | TOPIQ↑ |
|---|---|---|---|---|---|---|---|---|
| StableSR-200 | 11.21 | 1928 | 0.3927 | 56.85 | 0.4272 | 0.6263 | 3.49 | 0.5818 |
| ResShift-15 | 0.93 | 174 | 0.2377 | 53.11 | 0.4168 | 0.5823 | 3.45 | 0.5877 |
| ResShift-4 | 0.37 | 174 | 0.2074 | 52.10 | 0.3888 | 0.5993 | 3.40 | 0.5679 |
| UPSR-5 | 0.35 | **122** | 0.2464 | 59.20 | 0.4695 | 0.6149 | 4.00 | 0.6491 |
| SinSR-1 | 0.12 | 174 | 0.2187 | 53.53 | 0.4152 | 0.6079 | 3.58 | 0.5950 |
| CTMSR-1 | **0.08** | 172 | 0.1969 | 60.17 | 0.4857 | 0.6912 | 4.08 | 0.6793 |
| GTASR-1 (Ours) | **0.08** | 172 | **0.1916** | **65.09** | **0.5826** | **0.7475** | **4.47** | **0.7423** |

*Table 7.* Quantitative comparison on RealSR with center-cropped $128 \times 128$ LR inputs. The best and second-best results are highlighted in **bold** and underline ("-N" behind the method name represents the number of inference steps).

| Methods | Runtime (s)↓ | Params (M)↓ | PSNR↑ | SSIM↑ | LPIPS↓ | CLIPIQA↑ | MUSIQ↑ | MANIQA↑ | NIQE↓ | LIQE↑ | TOPIQ↑ |
|---|---|---|---|---|---|---|---|---|---|---|---|
| AddSR-4 | 0.70 | 2280 | 24.23 | 0.7032 | 0.3141 | 0.5127 | 62.64 | 0.4116 | 6.19 | 3.38 | 0.5535 |
| OSEDiff-1 | 0.23 | 1775 | 25.15 | 0.7341 | **0.2921** | 0.6682 | **69.08** | 0.4717 | 5.64 | **4.06** | 0.6253 |
| GTASR-1 (Ours) | **0.08** | **172** | **25.92** | **0.7471** | 0.3041 | **0.6962** | 67.31 | **0.4930** | **5.33** | 3.70 | **0.6780** |

*Table 8.* Quantitative results of models on two real-world datasets. The best and second best results are highlighted in **bold** and underline.

| Methods | RealLQ250 | | | | | | RealSet65 | | | | | |
|---|---|---|---|---|---|---|---|---|---|---|---|---|
| | CLIPIQA↑ | MUSIQ↑ | MANIQA↑ | NIQE↓ | LIQE↑ | TOPIQ↑ | CLIPIQA↑ | MUSIQ↑ | MANIQA↑ | NIQE↓ | LIQE↑ | TOPIQ↑ |
| AddSR-4 | 0.5732 | 64.40 | 0.3681 | 4.44 | 3.26 | 0.5486 | 0.5917 | 63.77 | 0.3903 | 5.22 | 3.49 | 0.5586 |
| OSEDiff-1 | 0.6722 | 69.556 | 0.4250 | **3.98** | **3.90** | 0.6075 | 0.7041 | 68.65 | 0.4699 | 4.79 | **4.08** | 0.6137 |
| GTASR-1 (Ours) | **0.7355** | **70.90** | **0.4870** | 4.32 | 3.79 | **0.7047** | **0.7491** | **69.49** | **0.4989** | **4.45** | 3.97 | **0.6937** |

## D.2. Comparison with T2I-based Distillation Methods

We further compare GTASR with representative T2I-based distillation methods, including AddSR (Tai et al., 2026) and OSEDiff (Wu et al., 2024a), under evaluation settings where a fair and well-defined comparison is possible. Since these latent-based models are trained and operate on $128 \times 128$ low-resolution inputs, a direct comparison on ImageNet with the standard $64 \times 64$ LR setting is not well-defined and would not yield a fair evaluation. Therefore, following prior work (Yang et al., 2024; Dong et al., 2025; Li et al., 2024), we omit comparisons on ImageNet-Test here. For the reference-based evaluation on RealSR as shown in Table 7, we follow established protocols in prior work (Duan et al., 2025; Wu et al., 2025b) by center-cropping the LR inputs to $128 \times 128$, ensuring compatibility with their VAE-based architectures and maintaining a consistent evaluation setup.

As evidenced in Tables 7 and 8, GTASR demonstrates a decisive advantage in the efficiency-performance trade-off. T2I-based methods rely on large-scale generative priors inherited from pretrained T2I models, which, while powerful, introduce substantial computational and parameter overhead; for instance, AddSR requires 2280 M parameters and 0.70s per inference. In stark contrast, GTASR runs $3\times$ to $9\times$ faster (0.08s) with only $\sim$10% of the parameters (172 M). Remarkably, this extreme efficiency does not compromise quality. GTASR achieves state-of-the-art fidelity (highest PSNR/SSIM) on RealSR and consistently outperforms the computationally expensive OSEDiff across most perceptual metrics on RealLQ250 and RealSet65. This validates that GTASR effectively achieves robust generative capabilities within a compact model, eliminating the need for cumbersome T2I architectures while delivering superior restoration quality.

It is particularly worth highlighting the disparity in training data scale. Representative T2I-based methods benefit significantly from powerful foundation models, such as Stable Diffusion 2.1 (Blattmann et al., 2023), which is pre-trained on the billion-scale LAION-5B (Schuhmann et al., 2022) dataset containing over 5.85 billion image-text pairs. In stark contrast, GTASR is trained from scratch solely on the ImageNet dataset, which consists of approximately 1.28 million samples—orders of magnitude fewer than the massive priors utilized by T2I models. The fact that GTASR achieves superior performance despite this significant scarcity of pre-trained priors strongly validates the effectiveness of our proposed consistency training

*Table 9.* Impact of the Trajectory Alignment (TA) strategy on restoration quality on ImageNet-Test. The best and second best results are highlighted in **bold** and underline.

| $\mathcal{L}_{\mathrm{CT}}$ | $\mathcal{L}_{\mathrm{TA}}$ | $\mathcal{L}_{\mathrm{DTM}}$ | $\mathcal{L}_{\mathrm{Stab}}$ | $\mathcal{L}_{\mathrm{Rect}}$ | PSNR↑ | SSIM↑ | LPIPS↓ | CLIPIQA↑ | MUSIQ↑ | MANIQA↑ | NIQE↓ | LIQE↑ | TOPIQ↑ |
|---|---|---|---|---|---|---|---|---|---|---|---|---|---|
| ✓ | | ✓ | ✓ | ✓ | **24.24** | **0.6540** | 0.2040 | 0.7036 | 61.94 | 0.4770 | 5.87 | 4.20 | 0.7036 |
| ✓ | ✓ | ✓ | ✓ | ✓ | 24.01 | 0.6520 | **0.1916** | **0.7475** | **65.09** | **0.5826** | **5.83** | **4.47** | **0.7423** |

paradigm. We hypothesize that given comparable data scale and quality, our method holds the potential for even more significant performance gains.

### D.3. Impact of Trajectory Alignment Strategy

To validate the necessity of our two-stage training strategy, we conduct an ablation study on the Trajectory Alignment (TA) loss, as reported in Table 9. Comparing the first two rows reveals a critical insight: removing $\mathcal{L}_{\mathrm{TA}}$ results in a significant performance drop across all perceptual metrics (e.g., MUSIQ decreases from 65.09 to 61.94, and MANIQA drops from 0.5826 to 0.4770).

This degradation underscores that $\mathcal{L}_{\mathrm{TA}}$ serves as the structural foundation for the entire framework. Without the explicit trajectory constraints imposed by $\mathcal{L}_{\mathrm{TA}}$ in stage I, the consistency learning process tends to drift into a "self-consistent" but erroneous manifold that deviates from the ground truth. Consequently, the subsequent gradient rectification losses $\mathcal{L}_{\mathrm{Stab}}$ and $\mathcal{L}_{\mathrm{Rect}}$ fail to unleash their full potential. Instead of refining the details of a correct trajectory, these losses are forced to compensate for fundamental structural errors, which exceeds their optimization capacity. Thus, the Trajectory Alignment strategy acts as a prerequisite anchor, ensuring that the model remains on the correct course for the subsequent fine-grained optimization to take effect.

### D.4. Additional Visual Comparison

We provide more visual examples of GTASR compared with recent state-of-the-art methods on ImageNet-Test and real-world datasets. The visual examples are shown in Figure 7, 8,9,10,11,12 and 13.

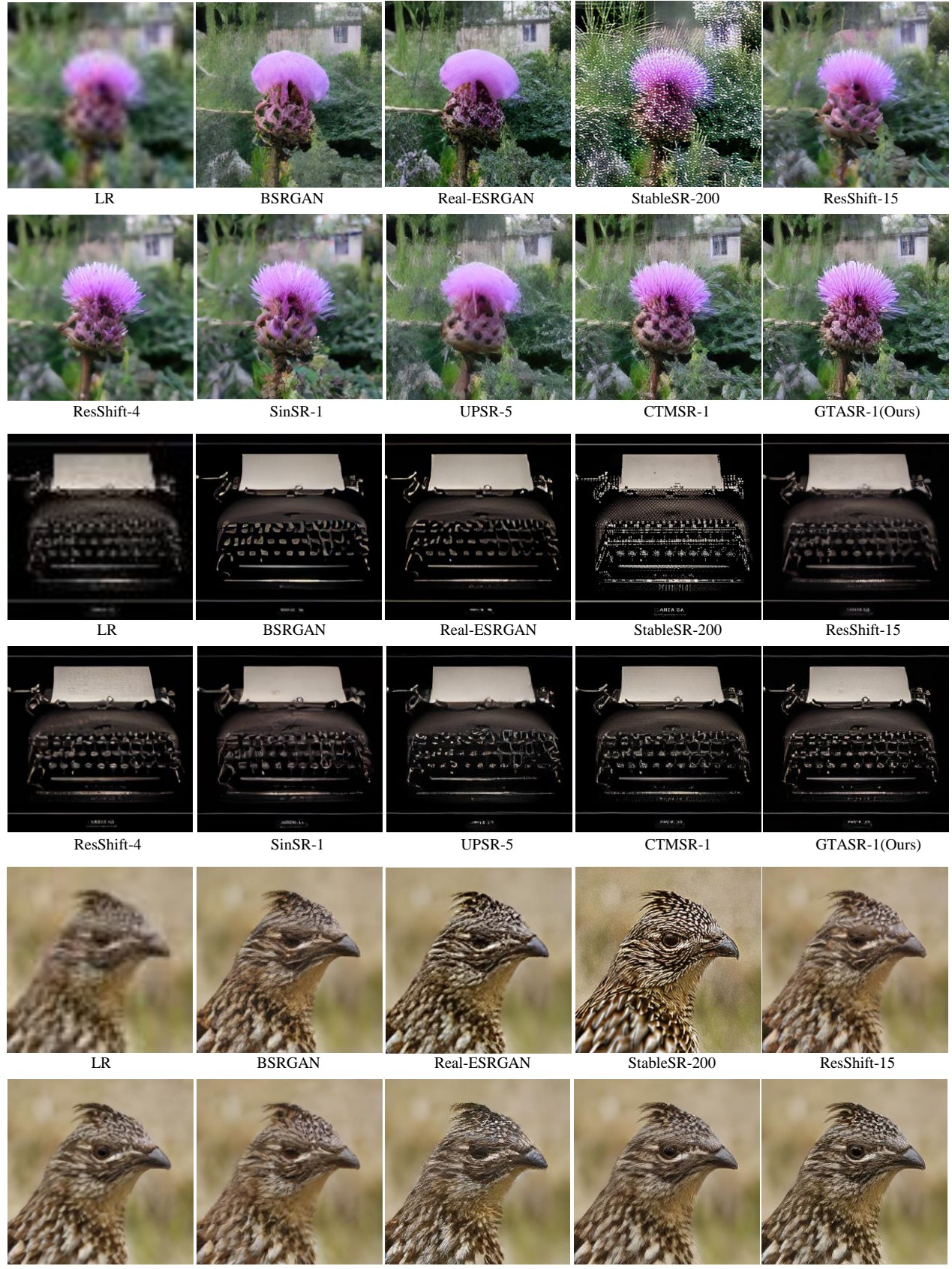

*Figure 7.* Visual comparison of different methods on a synthetic dataset. Please zoom in for more details.

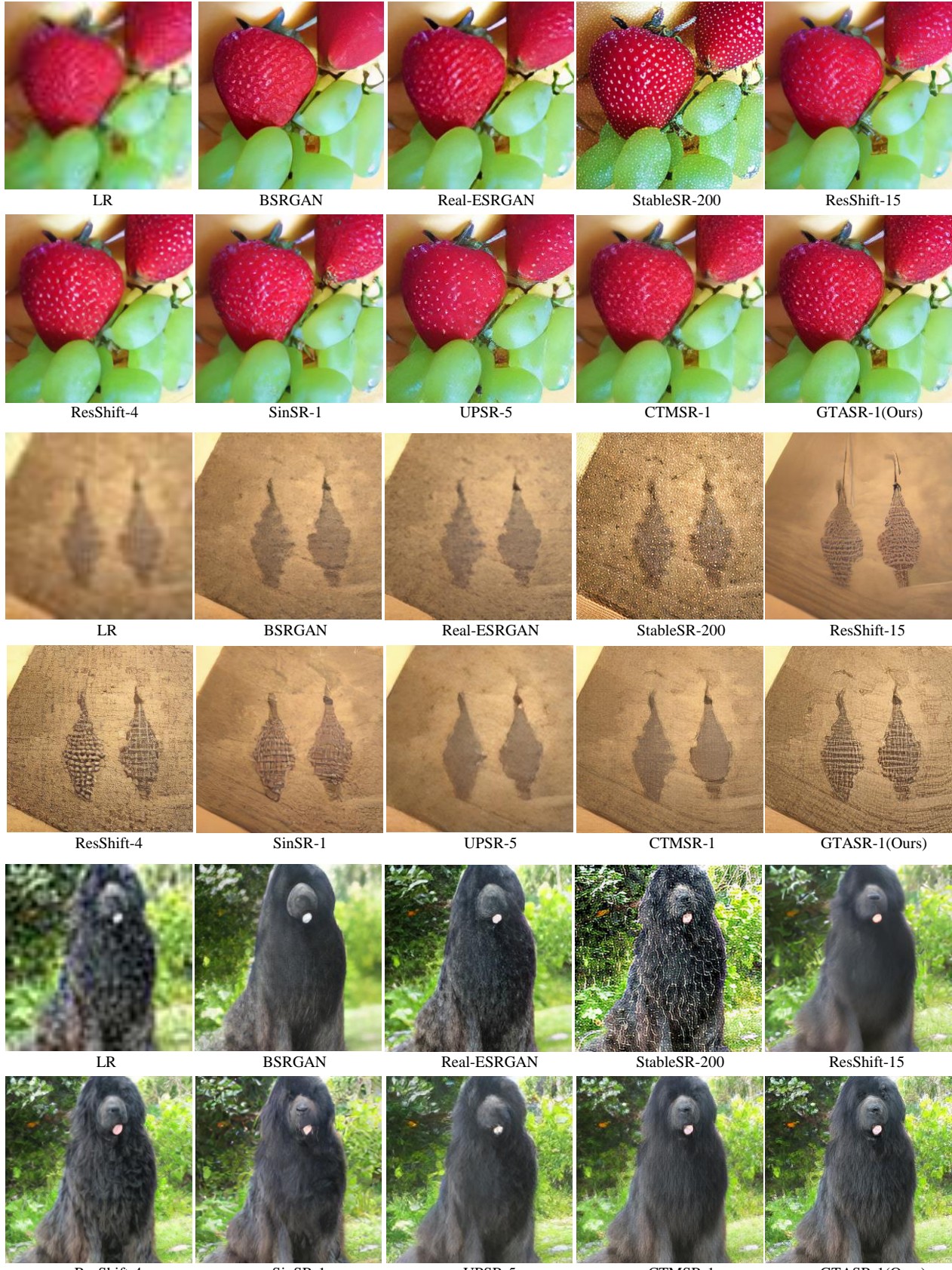

*Figure 8.* Visual comparison of different methods on a synthetic dataset. Please zoom in for more details.

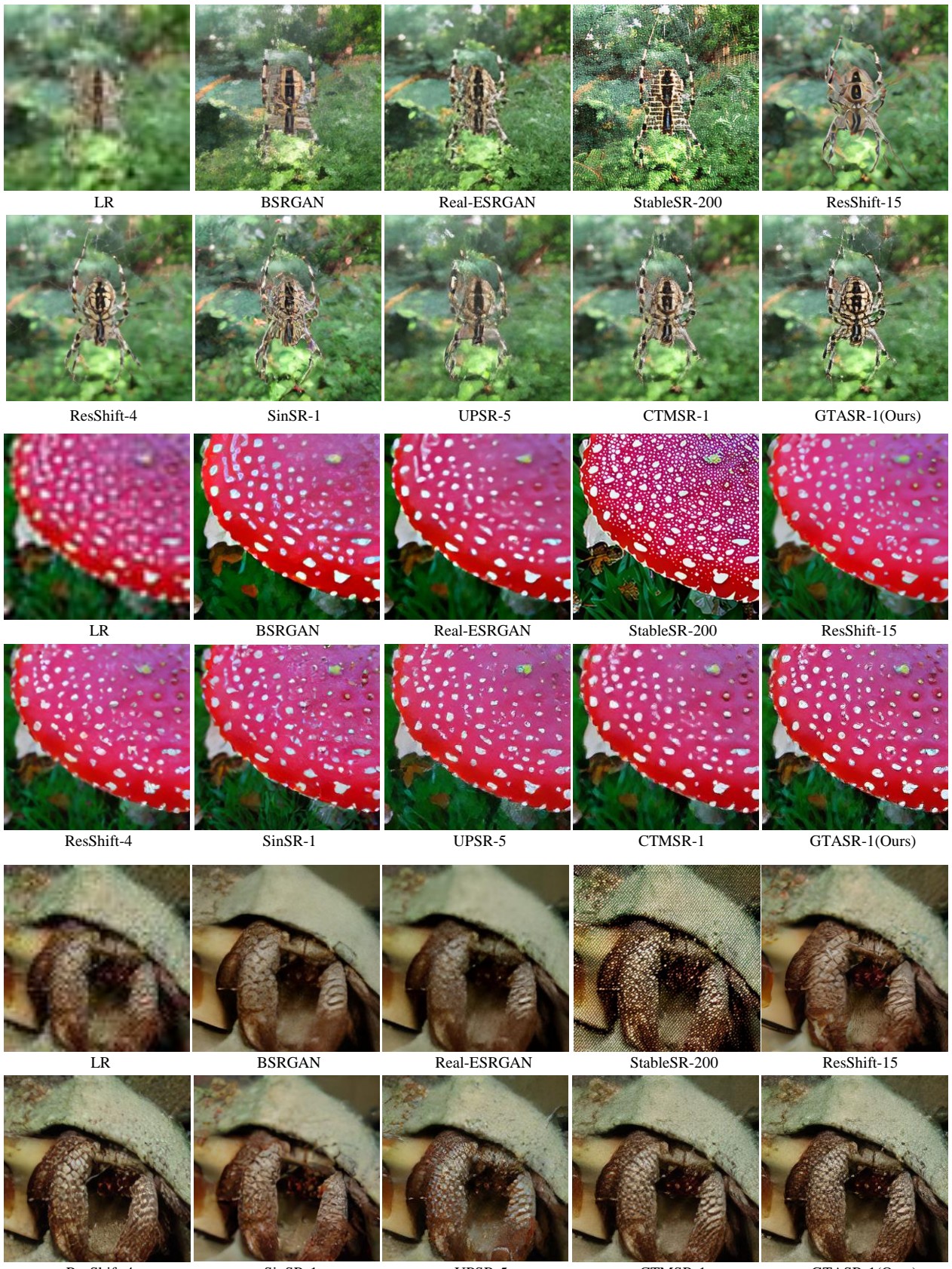

*Figure 9.* Visual comparison of different methods on a synthetic dataset. Please zoom in for more details.

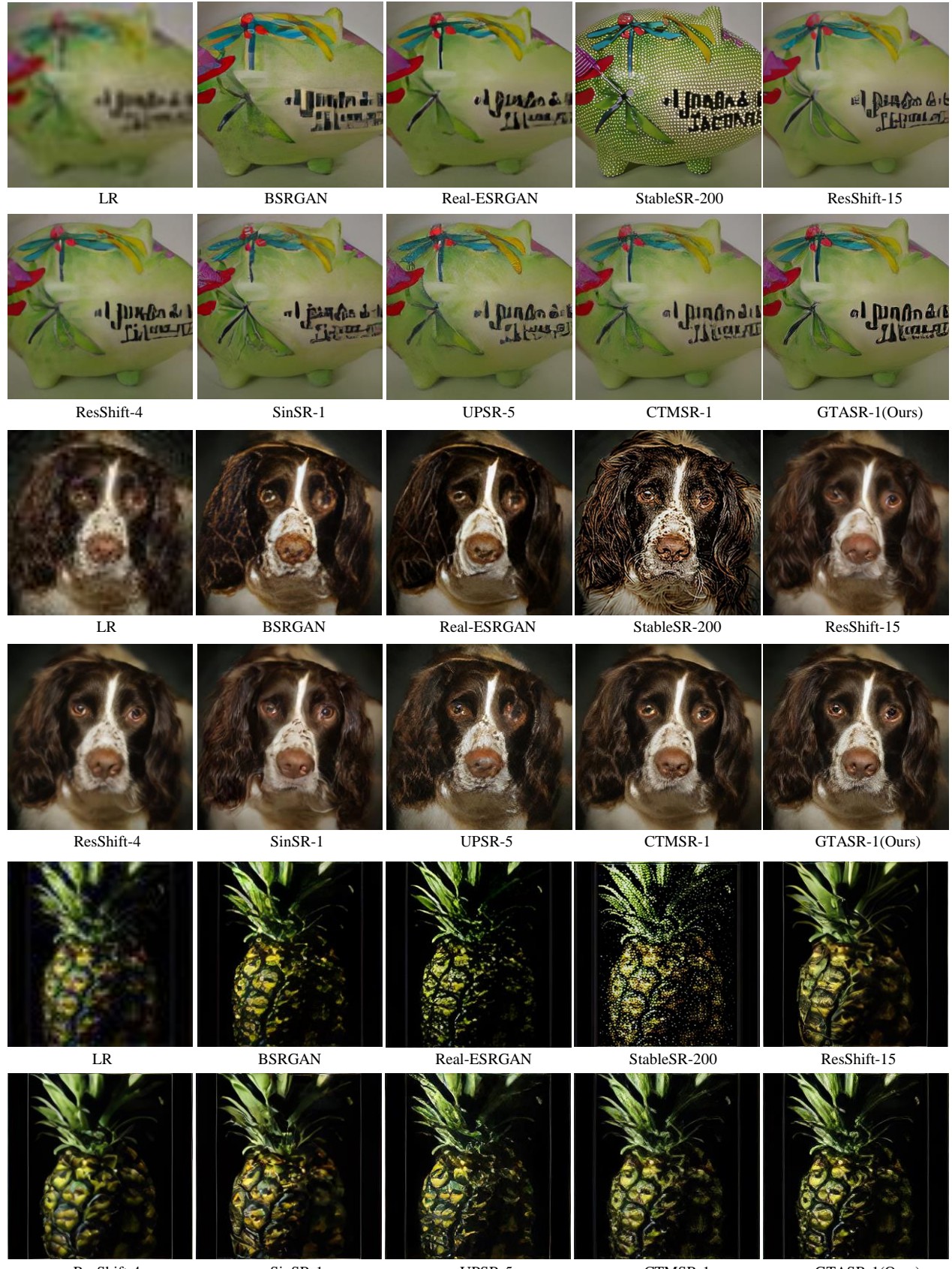

*Figure 10.* Visual comparison of different methods on a synthetic dataset. Please zoom in for more details.

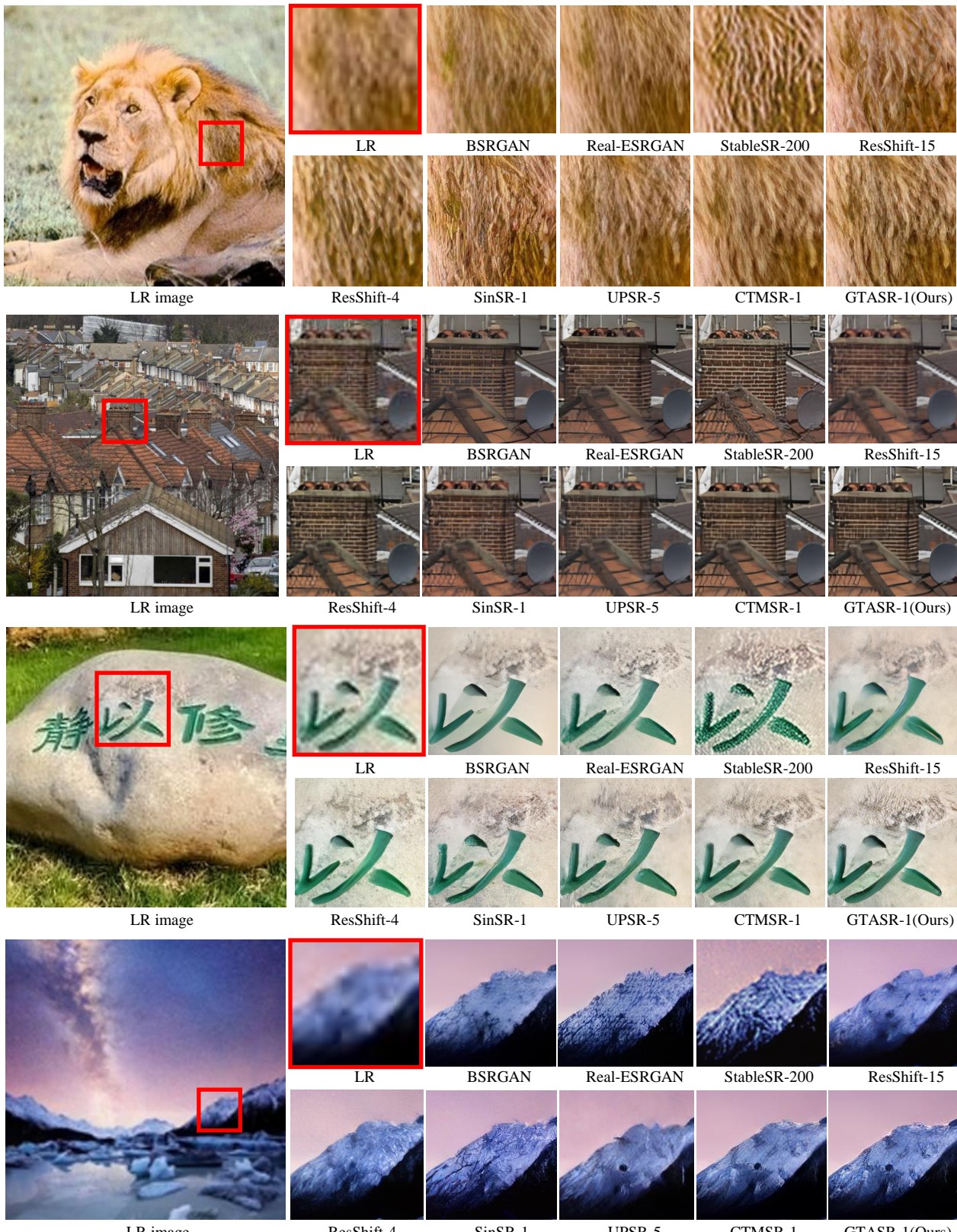

*Figure 11.* Visual comparison of different methods on real-world datasets. Please zoom in for more details.

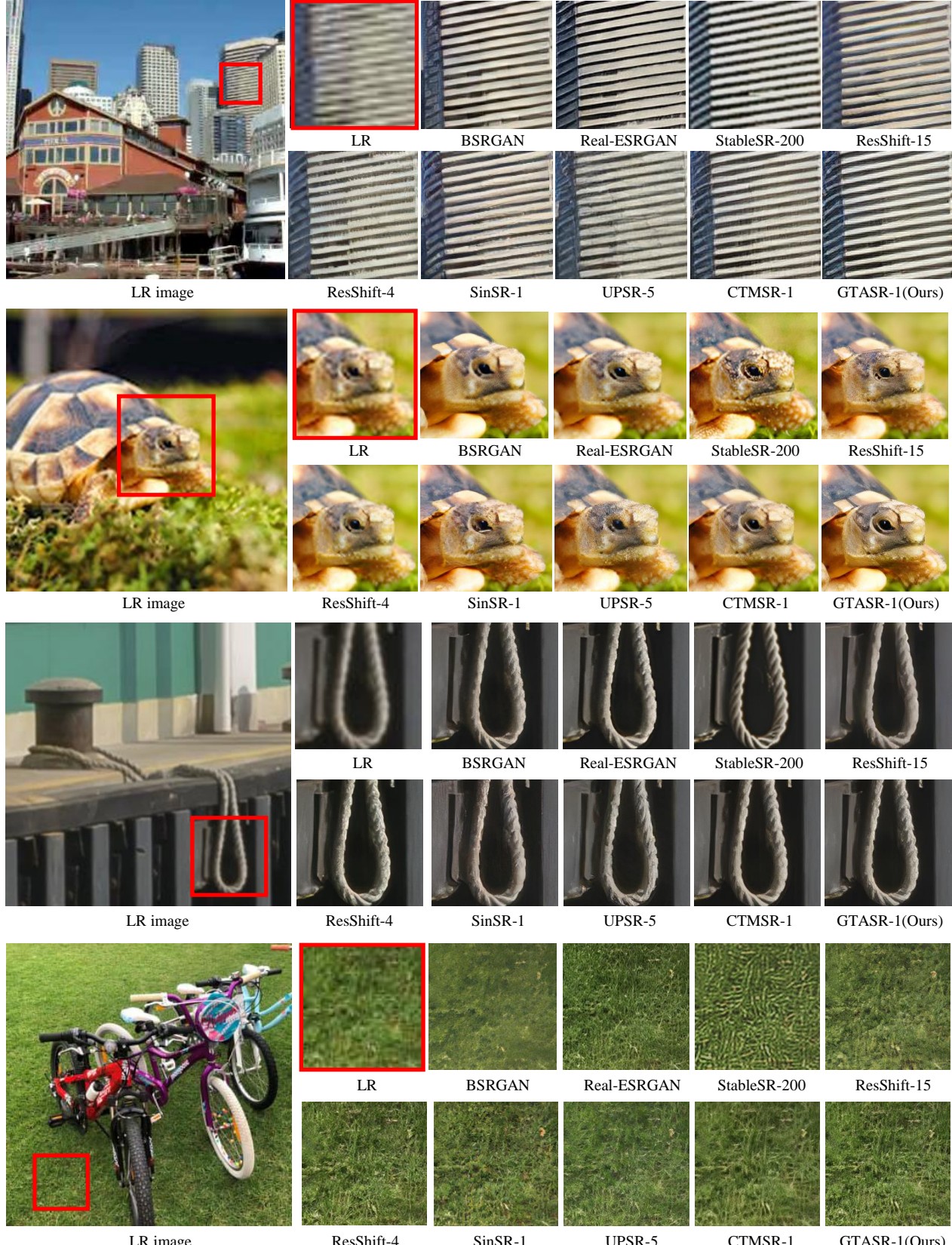

*Figure 12.* Visual comparison of different methods on real-world datasets. Please zoom in for more details.

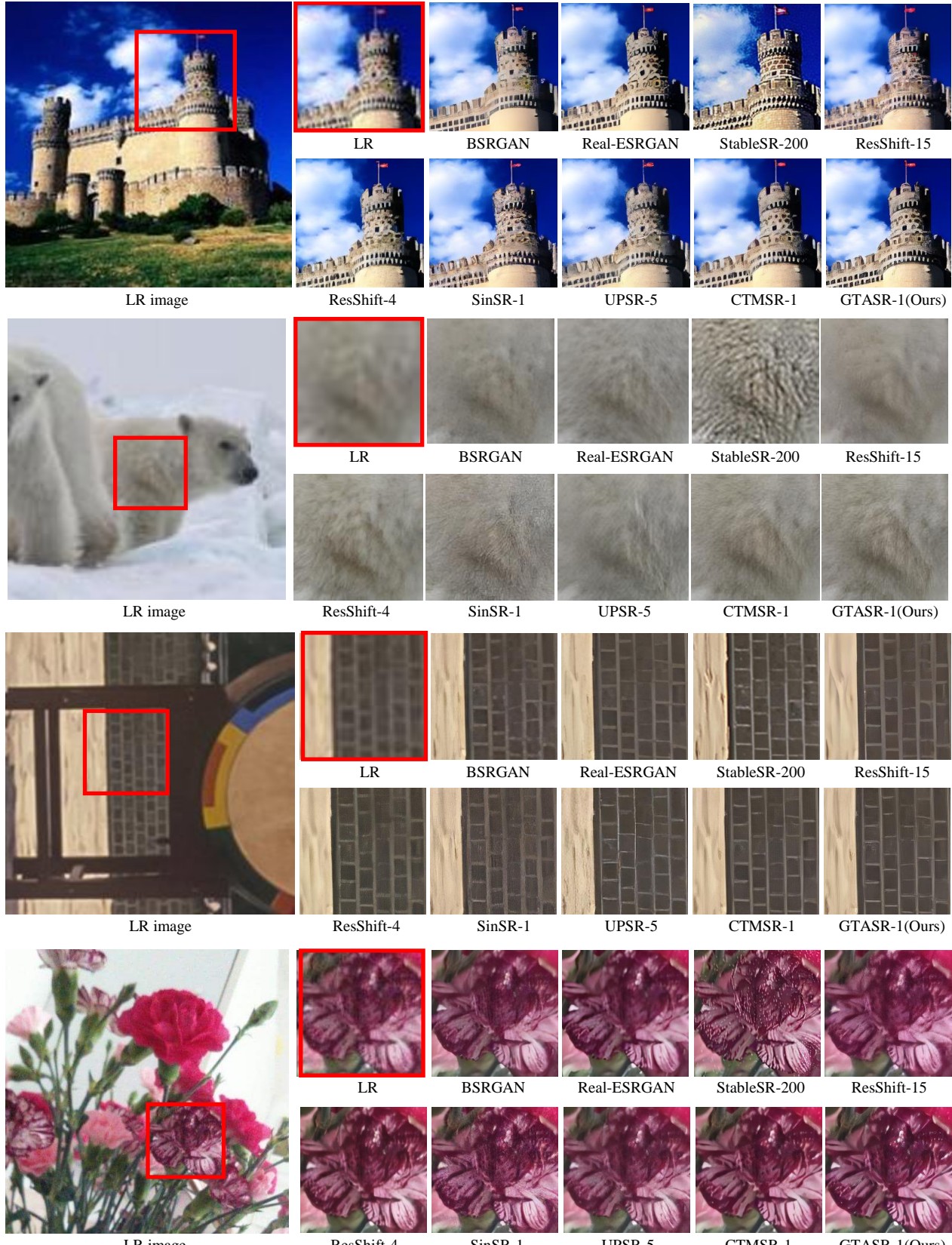

*Figure 13.* Visual comparison of different methods on real-world datasets. Please zoom in for more details.

