# OpenReview forum: "Joint Geometric and Trajectory Consistency  Learning for One-Step Real-World Super-Resolution"
_ICML.cc/2026/Conference — ICML 2026 regular_

### Official Review · Reviewer_rdTA · 2026-02-26

**Soundness:** 3
**Presentation:** 3
**Significance:** 3
**Originality:** 3
**Overall Recommendation:** 4
**Confidence:** 4

**Summary:**

This paper proposes GTASR, a one-step real-world super-resolution method based on consistency training. It identifies two issues in prior one-step consistency SR (e.g., drift along the learned trajectory and geometric/structural decoupling), and addresses them with (i) a full-path projection–based trajectory alignment loss to reduce drift, and (ii) a dual-reference structural rectification scheme that enforces Sobel-domain structural consistency and GT rectification. Experiments on synthetic paired and real-world benchmarks report improved perceptual/NR-IQA metrics while keeping one-step inference speed.

**Compliance With Llm Reviewing Policy:**

Affirmed.

**Final Justification:**

The author has addressed my concerns, so I will maintain my score.

**Key Questions For Authors:**

1. Can you provide an ablation that replaces LPIPS in the trajectory alignment loss with (a) only L1/Charbonnier and (b) alternative perceptual features, and report how sensitive the gains are to this choice?
2. What is the incremental benefit of each structural design choice (Sobel operator, dual-reference formulation, stop-gradient stabilization) compared to a strong “edge/gradient loss” baseline added to CTMSR under identical training settings?
3. Can you provide more failure cases on real, unpaired inputs and analyze typical artifacts (e.g., over-texturing, edge ringing, structural hallucination) along with guidance on when your method should not be used?

**Limitations:**

yes

**Strengths And Weaknesses:**

Strengths
1. Tackles an important practical goal: fast one-step real-world SR without relying on large teacher diffusion models.
2. Clear problem diagnosis (trajectory drift and geometric decoupling) and a method that directly targets both.

Weaknesses
1. The novelty of the structural component may be incremental, because the proposed rectification largely resembles adding an edge-domain (Sobel) constraint, and it is not fully established that this is qualitatively different from prior “edge/gradient loss” practices beyond the specific stop-gradient formulation.
2. The paper would benefit from a deeper failure-case analysis showing when the method introduces over-sharpening, hallucinated textures, or structural artifacts, especially on truly unpaired real images.
3. The trajectory alignment loss appears to depend heavily on perceptual distances (e.g., LPIPS) to avoid “noise cancellation,” but the paper does not convincingly show how robust the approach is when using purely pixel-wise distances or alternative perceptual metrics.

---

> ### Author Rebuttal · Authors · 2026-03-31
>
> We thank Reviewer rdTA for the positive feedback and constructive suggestions that contribute to improving this paper. Below, we address your comments point by point.
>
> **W1 & Question2: Edge loss baseline vs component-wise contributions**
>
> **A1:** Thank you for the comment. We clarify that the proposed structural component **should not be interpreted as simply adding an edge/gradient loss on top of CTMSR**, but instead arises from a unified modeling of structural error.
>
> Specifically, we define the structural error in Eq. (29) using a Sobel operator and expand it along the fake and ideal trajectories. By applying the triangle inequality, we derive an upper bound (Appendix B.3), which naturally decomposes into two components: target bias and consistency gap. Under this result, $\mathcal{L}_{Rect}$ corresponds to the target bias term in the structural space, and thus takes a form equivalent to a standard Sobel-based edge/gradient loss. Importantly, **this equivalence is not by design, but arises as a direct consequence of the derivation**. The second term governs structural consistency along the trajectory. This implies that both constraints arise from the same decomposition and must be considered jointly, rather than being introduced incrementally as extensions of a single edge loss.
>
> This is also supported by empirical evidence. As shown in Table 5, under the same training setting, using $\mathcal{L}_{Rect}$ alone (i.e., the same form as a standard edge loss) improves MUSIQ from 58.91 to 60.41, while the full formulation further improves it to 65.09. This shows that **the gain does not stem from the Sobel operator itself**, and more importantly, that **the joint constraint is substantially more effective than a naive edge loss**, as it is derived from **a principled decomposition rather than heuristic regularization**.
>
> Stop-gradient stabilization follows OSEDiff (VSD) and CTMSR (DTM). It fixes the reference branch to avoid the **“moving target”** problem. In Eq. (13), without stop-gradient, the reference trajectory would be updated jointly, causing the optimization target to change during training and weakening the constraint signal.
>
> **W2 & Question3: Failure cases and artifact analysis**
>
> **A2:** Thanks for your constructive suggestion. We agree that analyzing failure cases on real unpaired inputs helps clarify the limitations of the method.
>
> **Edge ringing** is not a primary issue in GTASR, with artifacts generally minimal. As noted, **structural hallucinations** may occur in some cases, which we attribute to the domain gap between synthetic training degradations and complex real-world inputs (out-of-domain). In such scenarios, even T2I-based methods may fail to recover accurate geometry, often producing perceptually plausible but structurally inconsistent results.
>
> While GTASR achieves the best performance among methods trained from scratch without T2I priors, it primarily enhances details via **sharpening** rather than generating rich textures. As a result, it may produce a less detailed appearance in complex regions (e.g., fur), while avoiding unrealistic **texture hallucinations**.
>
> We will further discuss these **limitations** in the revision: structural hallucinations caused by domain gap, and less texture-rich details due to the sharpening-oriented behavior of GTASR. Corresponding visual examples are available at: https://figshare.com/s/faa5811b42ef704738ec.
>
> **W3 & Question1: Sensitivity to perceptual loss choice**
>
> **A3:** Thank you for the valuable suggestion. We evaluate the sensitivity of TA to the choice of distance metric by replacing LPIPS with Charbonnier and DISTS while keeping all other settings fixed as follows:
>
> Distance in $\mathcal{L}_{TA}$|PSNR↑|LPIPS↓|CLIPIQA↑|MUSIQ↑|MANIQA↑|LIQE↑|TOPIQ↑
> -|-|-|-|-|-|-|-
> w/o $\mathcal{L}_{TA}$|24.65|0.2073|0.6060|55.38|0.3830|3.86|0.5939
> Charbonnier|**25.28**|0.2014|0.6158|56.71|0.4176|3.91|0.6114
> DISTS+Charbonnier|25.13|0.1998|0.6197|57.06|0.4228|3.95|0.6268
> LPIPS+Charbonnier(Ours)|25.05|**0.1856**|**0.6291**|**58.40**|**0.4457**|**4.05**|**0.6433**
>
> Charbonnier alone already improves over w/o $\mathcal{L}_{TA}$, **demonstrating that TA does not rely on LPIPS to be effective**. However, its gain is limited because, under shared-noise projections, pixel-wise distances cancel out the noise component and reduce the supervision to a scaled clean-space constraint, where variation across timesteps is mainly governed by a scaling factor rather than noise-dependent differences, resulting in weaker discriminative guidance.
>
> Replacing it with DISTS further improves perceptual quality, indicating that nonlinear feature representations prevent the supervision from degenerating into a pixel-wise constraint and instead preserve trajectory-level guidance. However, it remains inferior to LPIPS (Ours), as DISTS emphasizes global texture statistics and is less sensitive to local spatial discrepancies, leading to weaker supervision for precise structural alignment.

---

> > ### Author Rebuttal · Reviewer_rdTA · 2026-04-03
> >
> > The author has addressed my concerns, so I will maintain my score.

---

> > > ### Author Response · Authors · 2026-04-03
> > >
> > > Dear Reviewer rdTA,
> > >
> > > Thank you for your timely feedback, constructive suggestions, and follow-up comment. We are pleased that our response has addressed your concerns, and we sincerely appreciate your valuable efforts in improving this work. We will further refine the manuscript in the revised version by incorporating your suggestions.

---

### Official Review · Reviewer_wdWi · 2026-03-12

**Soundness:** 3
**Presentation:** 3
**Significance:** 3
**Originality:** 2
**Overall Recommendation:** 4
**Confidence:** 3

**Summary:**

This paper presents GTASR, a lightweight consistency training framework tailored for real world image super resolution that bypasses the high computational costs of iterative diffusion models and the parameter bloat of distillation methods. To overcome the inherent flaws of standard consistency models, namely consistency drift and geometric decoupling, the authors propose a Trajectory Alignment strategy to correct the tangent vector field via full path projection, paired with a Dual Reference Structural Rectification mechanism to enforce strict structural coherence. As a result, GTASR achieves outstanding visual quality and outperforms representative baselines while maintaining minimal inference latency.

**Compliance With Llm Reviewing Policy:**

Affirmed.

**Key Questions For Authors:**

Please see weakness

**Strengths And Weaknesses:**

Pros:
1. Overall, this paper is well-written and well-organized.
2. The motivation for proposing the method is clear, and it demonstrates certain performance improvements.
3. Performance experiments and ablation experiments are extensive.

Cons:
1. Compared to CTMSR, the contribution of this work is incremental.
2. Based on the ablation experiment, the improvement provided by L_Rect is quite limited.
3. Two-stage training involves additional time and computational overhead. It is recommended to compare the training costs of different methods.

---

> ### Author Rebuttal · Authors · 2026-03-30
>
> We thank Reviewer wdWi for the positive feedback and constructive suggestions that contribute to improving this paper. Below, we address your comments point by point.
>
> **C1: Limited Novelty over CTMSR**
>
> **A1:** Thank you for the comment. We clarify that GTASR is motivated by two observed failure modes in consistency-based training, namely trajectory drift and geometric decoupling, and introduce mechanism-level improvements grounded in training dynamics.
>
> For TA loss, the key lies in **aligning supervision with the time-dependent noise structure**. CTMSR training mainly relies on the CT loss, whose supervision is defined along recursively generated trajectories; once intermediate states deviate, the supervision target shifts accordingly and errors accumulate, leading to trajectory drift. In contrast, the TP loss enforces a uniform clean-target constraint across all timesteps $t$, ignoring the variation in noise levels. Instead, the TA loss re-projects predictions to the corresponding noisy states, so that supervision matches the signal-to-noise ratio at each timestep—guiding global evolution at high noise and refining details at low noise. This yields a **time-dependent, frequency-adaptive supervision** that more properly rectifies the tangent vector field and mitigates drift. Such behavior **is difficult to achieve by uniformly applying constraints or simply adding losses across timesteps**.
>
> For DRSR, the key is that the constraints are **not heuristically introduced**. As shown in Appendix B.3, we define the structural error in Eq. (29) and derive its upper bound via the triangle inequality, which naturally decomposes into consistency gap and target bias. Accordingly, the Stab loss and Rect loss are introduced to suppress these two terms, rather than being ad-hoc losses. This decomposition reveals a **missing component in DTM**: the original objective does not jointly constrain these two error terms. Therefore, DRSR does not merely strengthen existing objectives, but explicitly incorporates these missing components into optimization via an upper-bound–guided formulation. This mechanism **is unlikely to be equivalently realized by simply stacking additional losses**.
>
> **C2: Effect of Rect Loss**
>
> **A2:** Thank you for your constructive question. We clarify that the Rect loss should **not be interpreted as an isolated add-on**, but as part of a unified formulation derived from structural error decomposition.
>
> Individually, it improves over the baseline (e.g., MANIQA 0.5070 → 0.5403, MUSIQ 58.91 → 60.41), confirming its effectiveness in reducing structural bias. However, this ablation is intended to validate the decomposition rather than assess each term in isolation.
>
> The Stab loss and Rect loss correspond to two coupled terms in the upper-bound analysis (consistency gap and target bias), **as derived in Appendix B**. These two terms arise from the same derivation, and the corresponding losses are designed accordingly, forming a unified constraint. They are inherently complementary, addressing different aspects of structural inconsistency—one enforces trajectory consistency, while the other corrects target bias. Using either term alone provides only partial supervision, while their joint optimization is **necessary to fully realize the constraint**, consistently yielding the best performance (Table 5 in our paper).
>
> **C3: Training cost comparison**
>
> **A3:** Thanks for your valuable suggestion. For a fair comparison, we re-trained CTMSR from scratch under the same settings and report all results in the following Table. Both methods adopt a two-stage training paradigm (all experiments conducted on NVIDIA GeForce RTX 4090 GPUs). Our method increases the total training time from 73.85h to 90.60h (≈+22.7%), while maintaining the **same inference time (0.08s)**.
>
> Method | Stage I (h)↓ | Stage II (h)↓ | Total (h)↓ | Infer (s)↓ | PSNR↑ | SSIM↑ | LPIPS↓ | CLIPIQA↑ | MUSIQ↑ | MANIQA↑ | NIQE↓ | LIQE↑ | TOPIQ↑
> ---|---|---|---|---|---|---|---|---|---|---|---|---|---
> **CTMSR-1 (Baseline)** | **73.1** | **0.753** | **73.85** | **0.08** | **24.79** | **0.6688** | 0.2041 | 0.6604 | 57.52 | 0.4317 | **5.87** | 3.96 | 0.6452
> **GTASR (Ours)** | 89.8 | 0.796 | 90.60 | **0.08** | 24.01 | 0.6520 | **0.1916** | **0.7475** | **65.09** | **0.5826** | 5.83 | **4.47** | **0.7423**
>
> In return, GTASR brings **clear perceptual gains**: LPIPS improves from 0.2041 to 0.1916, CLIPIQA from 0.6604 to 0.7475, MUSIQ from 57.52 to 65.09, MANIQA from 0.4317 to 0.5826, and TOPIQ from 0.6452 to 0.7423. Although PSNR/SSIM decreases slightly, this reflects a common perception–distortion trade-off in real-world SR, where perceptual quality is more relevant to visual realism than distortion metrics alone.
>
> Overall, the extra training cost is **moderate**, incurred only offline, and yields **substantially better perceptual results without any inference-time penalty**. We will include a comparison of training costs in the revised manuscript.

---

> > ### Author Rebuttal · Reviewer_wdWi · 2026-04-01
> >
> > Thank you for your response. I still have some concerns about novelty, so I’d like to keep the original weak accept score.

---

> > > ### Author Response · Authors · 2026-04-02
> > >
> > > Dear Reviewer wdWi,
> > >
> > > Thank you for your timely feedback, constructive suggestions, and follow-up comment. We understand the remaining concerns regarding novelty. We would like to further clarify the key insight behind GTASR, which addresses a limitation that is not explicitly handled by existing consistency- or trajectory-matching-based formulations.
> > >
> > > Existing consistency-based SR methods implicitly assume that enforcing consistency in function space is sufficient to ensure correct trajectory evolution. Our key observation is that this assumption does not necessarily hold: a model can satisfy consistency constraints while still evolving along an incorrect trajectory.
> > >
> > > GTASR is designed to resolve this gap by explicitly modeling **trajectory correctness** and its interaction with structural geometry.
> > >
> > > **(1) From enforcing consistency to modeling trajectory correctness**
> > >
> > > Prior approaches impose point-wise constraints (e.g., adjacent-state consistency or clean target regression), which restrict outputs but leave the trajectory dynamics underdetermined. As a result, models may satisfy consistency while deviating from the correct evolution path.
> > >
> > > GTASR instead supervises the **geometry of the PF-ODE trajectory**. TA operates on the noisy manifold and aligns supervision with the time-dependent diffusion process, thereby correcting the tangent vector field. This shifts the objective from fitting outputs to controlling trajectory evolution. This issue is difficult to address with existing point-wise objectives, since they do not directly constrain the trajectory itself. This directly leads to improved structural fidelity in practice, avoiding artifacts such as distorted edges and detail collapse observed in baseline methods.
> > >
> > > **(2) A limitation of trajectory matching: geometric decoupling**
> > >
> > > We further show that trajectory matching objectives such as DTM may still be insufficient to ensure structural consistency. Specifically, we identify **geometric decoupling**, where perceptual alignment improves while local geometric structures diverge.
> > >
> > > This reflects a fundamental limitation of perceptual objectives, which are insensitive to local geometric orientation and thus fail to enforce structural consistency. This explains the structural inconsistencies (e.g., misaligned textures and unstable edges) observed in existing methods.
> > >
> > > **(3) From error decomposition to principled objective design**
> > >
> > > We address this issue through a **theoretically grounded formulation** rather than a heuristic design.
> > >
> > > Starting from a rigorous analysis of trajectory-induced structural error, we derive that the accumulated discrepancy admits an upper-bound decomposition into two terms: a **consistency gap** and a **target bias**. This decomposition establishes a direct link between trajectory dynamics and structural error, and directly determines the form of the corresponding optimization objectives.
> > >
> > > Based on this result, DRSR is constructed by explicitly optimizing these two components, yielding two objectives that target the sources of structural inconsistency. As a result, the losses in DRSR are not introduced heuristically, but arise naturally from the theoretical formulation.
> > >
> > > **(4) A coupled dynamical system perspective**
> > >
> > > From a broader perspective, GTASR can be viewed as a **coupled dynamical system**, where trajectory evolution and geometric consistency are jointly modeled.
> > >
> > > TA governs how the trajectory evolves, while DRSR constrains where it converges. This coupling resolves the under-constrained nature of prior formulations and leads to stable and structurally coherent solutions.
> > >
> > > In summary, GTASR is built upon a **new perspective** that consistency alone does not guarantee correct trajectory evolution. By explicitly modeling trajectory behavior together with structural constraints, it introduces a formulation that cannot be achieved by existing objectives, which is also reflected in the improved structural consistency and visual quality observed in our results. We hope this clarification helps resolve the remaining concerns regarding novelty and better reflects the contribution of our work.

---

### Official Review · Reviewer_2zXm · 2026-03-13

**Soundness:** 3
**Presentation:** 4
**Significance:** 3
**Originality:** 3
**Overall Recommendation:** 5
**Confidence:** 4

**Summary:**

This paper focuses on one-step real-world image super-resolution using consistency training, with the goal of improving both efficiency and structural fidelity relative to prior consistency-based SR methods such as CTMSR. The authors identify two issues in prior work: consistency drift from transitive consistency supervision and what they call geometric decoupling, where perceptual or pixel alignment does not guarantee structural coherence. To address these, they propose GTASR, which combines a Trajectory Alignment (TA) loss in Stage I and a Dual-Reference Structural Rectification (DRSR) mechanism in Stage II, built on top of consistency training and distribution trajectory matching. Experiments on synthetic and real-world SR benchmarks report improved perceptual quality and similar one-step runtime compared with representative baselines.

**Compliance With Llm Reviewing Policy:**

Affirmed.

**Final Justification:**

The rebuttal has addressed my main concerns, and the author' responding is fast and effective. I will raise my score from 4 to 5.

**Key Questions For Authors:**

Please refer to the **Weaknesses and Questions**

**Limitations:**

Please describe the ability of on-device usage, like mobile.

**Strengths And Weaknesses:**

**Strengths**

1. The method is reasonably well motivated at a high level. The split into two issues, trajectory drift and structural inconsistency, gives the paper a coherent narrative, and Figure 1 helps communicate the two-stage training pipeline and the role of the online, reference, and target networks.
2. The ablation study is informative and generally supports the role of the proposed modules. Table 4 shows that TA is already beneficial in Stage I, while Table 5 suggests that the Stage II losses are complementary, with the full model producing the best perceptual metrics among the listed variants.

**Weaknesses and Questions**
1. The method appears to be an incremental extension of CTMSR rather than a fundamentally new framework. TA is essentially re-projection-based supervision on the noisy manifold, and DRSR adds Sobel-based structural losses on top of DTM.
2. The proposed notion of geometric decoupling is not yet convincing and may even seem counter to the usual intuition that diffusion-based methods emphasize semantic/perceptual information over strict pixel alignment.
3. The problem statement in the introduction is imprecise: the first sentence describes a broader image restoration problem rather than super-resolution in the strict sense.
4. The paper lacks a more complete computational efficiency analysis, especially FLOPs/MACs.
5. Comparisons with several recent relevant methods are missing, including PiSA-SR (CVPR 25), TSD-SR (CVPR 25), OSEDiff (NeurIPS 24), InvSR (CVPR 25), HYPIR (SIGGRAPH Asia 25).

---

> ### Author Rebuttal · Authors · 2026-03-30
>
> We thank Reviewer 2zXm for the positive feedback and constructive suggestions that contribute to improving this paper. Below, we address your comments point by point.
>
> **W1: Perceived Incremental  extension of CTMSR**
>
> **A1:** Thank you for the insightful comment. We respectfully clarify that GTASR is motivated by two fundamental limitations in consistency-based training—trajectory drift and geometric decoupling—and introduces **principled designs** to address them.
>
> Conventional CT loss lacks explicit vector field guidance, while naive TP loss applies uniform endpoint anchoring across timesteps and remains essentially point-to-point regression. In contrast, TA loss introduces **time-dependent supervision** that explicitly rectifies the tangent vector field, enabling accurate trajectory evolution beyond uniform constraints, thereby constituting a shift rather than an incremental modification of existing losses.
>
> The geometric decoupling indicates that DTM-style distribution alignment is **insufficient to constrain local geometry**. Based on our **analytical derivation (Appendix B)**, we decompose the structural inconsistency into consistency gap and target bias, and design Stab loss and Rect loss to constrain them, yielding a dual-reference mechanism rather than a heuristic edge-based regularization.
>
> **W2: Doubts on Geometric Decoupling**
>
> **A2:** Thank you for the insightful comment. We agree that diffusion models prioritize perceptual quality over strict pixel alignment. However, we clarify that DTM relies on perceptual metrics (e.g., LPIPS), whose features are spatially invariant with large receptive fields, making them **insensitive to local geometric variations**, thereby leaving geometry under-constrained.
>
> Therefore, we do not aim for strict pixel alignment, but emphasize that **structural instability degrades perceptual quality**, as it leads to misaligned edges and spatially inconsistent textures that violate natural image structures. This is further supported by Table 5 in our paper, where removing DRSR leads to clear drops, e.g., CLIPIQA (0.7475 → 0.6965), MANIQA (0.5826 → 0.5070), and MUSIQ (65.09 → 58.91). We will further clarify this causal relationship in the revision to avoid it being misinterpreted as introducing an additional geometric objective.
>
> **W3: Imprecise Problem Statement**
>
> **A3:** Thank you for your constructive suggestion. To strictly define the scope of super-resolution, we have revised the opening sentence of the Introduction as follows:
>
> *Image Super-Resolution (ISR) aims to reconstruct a High-Resolution (HR) image from its corresponding degraded Low-Resolution (LR) input.*
>
> **W4: Incomplete Efficiency Analysis**
>
> **A4:** Thank you for your valuable suggestion. We will add the following MACs comparison to our paper for the $128\times128\to512\times512$ scale:
>
> Methods|StableSR|ResShift-15|ResShift-4|SinSR|UPSR|CTMSR|GTASR(Ours)
> -|-|-|-|-|-|-|-
> MACs(G)↓|79940|5491|3034|2649|813|**610**|**610**
>
> **W5: Lack of Recent Comparisons**
>
> **A5:** Thank you for pointing out these excellent works. For fair comparison, we primarily focus on models that, like our GTASR, are **trained from scratch**, such as CTMSR (ICCV' 25) and UPSR (CVPR' 25). Comparisons and analyses with representative methods leveraging strong **T2I priors** (e.g., OSEDiff, AddSR) are provided in **Appendix D.2**. To address your concerns, we evaluate the specific T2I-based methods you mentioned on RealSR below. Notably, our method achieves competitive performance **without relying on strong T2I priors, while remaining lightweight and efficient at inference**.
>
> Metric|PiSA-SR|TSD-SR|OSEDiff|InvSR|HYPIR|GTASR(Ours)
> -|-|-|-|-|-|-
> Runtime (s)↓|0.51|0.39|0.43|*0.35*|0.65|**0.08**
> Params (B)↓|1.78|2.11|1.78|*1.33*|2.08|**0.17**
> PSNR↑|*25.50*|24.81|25.15|24.30|22.89|**25.92**
> CLIPIQA↑|0.6699|**0.7160**|0.6682|*0.6785*|0.6512|*0.6962*
> MUSIQ↑|*70.15*|**71.19**|69.08|67.31|66.42|67.31
> MANIQA↑|0.4917|0.4733|0.4717|*0.4927*|0.4896|**0.4930**
> NIQE↓|*5.51*|**5.13**|5.64|5.62|5.50|*5.33*
>
> **Limitation1: On-Device Deployment Feasibility**
>
> **A6:**  Thank you for your constructive feedback. We adopt a simplified on-device validation pipeline by exporting the model to a fixed-input ONNX format, configuring the runtime environment in Xcode, and evaluating it on a mobile device (iPhone 15 Pro Max).
>
> For the 128→512 super-resolution task, GTASR runs in 7.92 s with 1.75 GB peak memory. Notably, even **without any pruning or quantization**, it remains stable on mobile, demonstrating strong deployment potential (demo: https://figshare.com/s/b20e63ac4c885ddbbcfe).
>
> We further evaluate representative methods under the same deployment pipeline. Only three methods are able to run successfully, among which GTASR and CTMSR achieve the best efficiency:
>
> Metric|StableSR|ResShift-15|ResShift-4|SinSR|UPSR|CTMSR|GTASR (Ours)
> -|-|-|-|-|-|-|-
> Time (s)↓|OOM|OOM|OOM|OOM|24.65|7.92|7.92
> Memory (GB)↓|OOM|OOM|OOM|OOM|2.44|1.75|1.75

---

> > ### Author Rebuttal · Reviewer_2zXm · 2026-04-02
> >
> > Thank you for the response.
> >
> > Some questions on your A5.
> >
> > The GTASR seems unbelievably fast compared to other methods, and thus, I question the experiment setting.
> >
> > For example, in the paper of HYPIR (Fig. 15), they test the speed of 1024 $\times$ 1024 image generation, which corresponds to 256 $\times$ 256 LR input. In their setting, they may cost thoroughly 4 times compared to yours. But they tested OSEDiff for 0.635s on RTX A6000, which means about 0.158s on 128$\times$128 input.
> >
> > RTX A6000 is slower than RTX 4090, especially in inference.
> >
> > The authors can refer to other **papers** to compare (OSEDiff, PiSA-SR, TSD-SR, InvSR, HYPIR, D3SR)
> >
> > I wonder whether the bit setting (FP32, FP16, BF16) is the same, and whether the test is conducted multiple times, getting an average, and whether the GPU is idle when testing. Maybe there exists some inconsistency in the settings.
> >
> > But overall, I will maintain my weak acceptance review.

---

> > > ### Author Response · Authors · 2026-04-03
> > >
> > > Dear Reviewer 2zXm,
> > >
> > > ***
> > > **[Update]** Thank you for raising the score! We appreciate your positive feedback. We will incorporate your suggestions to further improve the clarity and completeness of the manuscript.
> > > ***
> > >
> > > Thank you for your timely feedback, constructive suggestions, and follow-up comment. We also apologize for the insufficient clarity in our previous rebuttal regarding the runtime evaluation. Following your suggestion, we revisited the runtime settings in related works (e.g., OSEDiff) and examined their official implementations. We find that OSEDiff reports **model forward latency**, while the T2I methods in our table **measure end-to-end inference time over 100 images on RealSR**, i.e., the full pipeline from data loading to saving (excluding model initialization and weight loading). These two protocols are fundamentally different and not directly comparable. We also confirm that **all methods follow their official precision settings and are evaluated with idle GPUs**.
> > >
> > > We re-evaluated the runtime under consistent settings. On an RTX 4090 (CPU: AMD EPYC 7352), using the official OSEDiff timing script, OSEDiff achieves 0.225s forward latency, while our method achieves 0.08s under the same protocol, demonstrating a clear efficiency advantage in the core inference stage. HYPIR reports results on RTX A6000; however, as we do not have access to this hardware, we are unable to reproduce its performance under the exact same setting, and we apologize for this limitation.
> > >
> > > We would like to clarify that **Table 6 in the main paper reports model forward latency**, and this setting itself is consistent with the official measurement protocol used in OSEDiff. The confusion arose in our previous rebuttal, where the T2I baselines were summarized using **end-to-end inference time**, while our method was reported with **model forward latency**, resulting in an inconsistent comparison. For fair comparison, we additionally report **end-to-end inference time**, where our method achieves 0.153s and OSEDiff 0.434s under the same setting.
> > >
> > > Overall, our method consistently demonstrates efficiency advantages under both metrics: forward latency (0.08s vs 0.225s) and end-to-end time (0.153s vs 0.434s). This gain comes from three aspects: **(i) compact model scale** (9.6% parameters, 26.9% MACs of OSEDiff), **(ii) lightweight architecture** without additional modules (e.g., DAPE), and **(iii) a streamlined pipeline**, where VAE decoding is removed in line with prior observations (e.g., UPSR), further reducing overhead in one-step settings.
> > >
> > > To further validate this, we report a unified comparison below, where our method achieves the **lowest runtime under both protocols** among all compared methods.
> > >
> > > Metric|PiSA-SR|TSD-SR|OSEDiff|InvSR|HYPIR|GTASR(Ours)
> > > -|-|-|-|-|-|-
> > > Model Forward Runtime (s)↓|0.25|0.20|0.23|*0.18*|0.36|**0.08**
> > > End-to-End Inference Time (s)↓|0.51|0.39|0.43|*0.35*|0.65|**0.15**
> > >
> > > We sincerely appreciate the reviewer for pointing out this issue. We acknowledge that our original presentation of the runtime evaluation was not sufficiently clear, which may have caused confusion. We will carefully revise the manuscript to provide a more explicit and accurate description of the runtime protocol, and update the corresponding statements accordingly to ensure clarity and avoid any ambiguity in the revised version. We hope this clarification adequately addresses your concerns.

---

### Decision · Program_Chairs · 2026-04-30

**Decision:**

Accept (regular)

**Comment:**

This paper develops a GTASR that contains a trajectory alignment strategy to rectify the tangent vector field via full-path projection, and a
dual-reference structural rectification mechanism to enforce strict structural constraints for efficient image super-resolution. The major concerns of reviewers include the limited novelty, insufficient evaluations about the efficiency of the proposed method and comparisons with related one-step diffusion-based methods, and limited performance improvement of the key component.

The provided rebuttal solves the concerns of reviewers. All reviewers are satisfied with the response.